# Evaluating Graph Neural Networks for Link Prediction: Current Pitfalls and New Benchmarking

**Juanhui Li**[1*], **Harry Shomer**[1*] , **Haitao Mao**[1] , **Shenglai Zeng**[1]
**Yao Ma**[2] , **Neil Shah**[3] , **Jiliang Tang**[1] , **Dawei Yin**[4]
[1]Michigan State University, [2]Rensselaer Polytechnic Institute
[3]Snap Inc., [4]Baidu Inc.
{lijuanh1,shomerha,haitaoma,zengshe1,tangjili}@msu.edu
may13@rpi.edu, nshah@snap.com, yindawei@acm.org

## Abstract

Link prediction attempts to predict whether an unseen edge exists based on only a portion of edges of a graph. A flurry of methods have been introduced in recent years that attempt to make use of graph neural networks (GNNs) for this task. Furthermore, new and diverse datasets have also been created to better evaluate the effectiveness of these new models. However, multiple pitfalls currently exist that hinder our ability to properly evaluate these new methods. These pitfalls mainly include: (1) Lower than actual performance on multiple baselines, (2) A lack of a unified data split and evaluation metric on some datasets, and (3) An unrealistic evaluation setting that uses easy negative samples. To overcome these challenges, we first conduct a fair comparison across prominent methods and datasets, utilizing the same dataset and hyperparameter search settings. We then create a more practical evaluation setting based on a **He**uristic **R**elated Sampling **T**echnique (HeaRT), which samples hard negative samples via multiple heuristics. The new evaluation setting helps promote new challenges and opportunities in link prediction by aligning the evaluation with real-world situations. Our implementation and data are available at `https://github.com/Juanhui28/HeaRT`.

## 1 Introduction

The task of link prediction is to determine the existence of an edge between two unconnected nodes in a graph. Existing link prediction algorithms attempt to estimate the proximity of different pairs of nodes in the graph, where node pairs with a higher proximity are more likely to interact [1]. Link prediction is applied in many different domains including social networks [2], biological networks [3], and recommender systems [4].

Graph neural networks (GNNs) [5] have gained prominence in recent years with many new frameworks being proposed for a variety of different tasks. Corresponding to the rise in popularity of GNNs, there has been a number of studies that attempt to critically examine the effectiveness of different GNNs on various tasks. This can be seen for the task of node classification [6], graph classification [7], knowledge graph completion (KGC) [8–10], and others [11].

However, despite a number of new GNN-based methods being proposed [12–15] for link prediction, there is currently no work that attempts to carefully examine recent advances in link prediction methods. Upon examination, we find that there are several pitfalls in regard to model evaluation that impede our ability to properly evaluate current methods. This includes:

---

*Equal contribution.

37th Conference on Neural Information Processing Systems (NeurIPS 2023) Track on Datasets and Benchmarks.

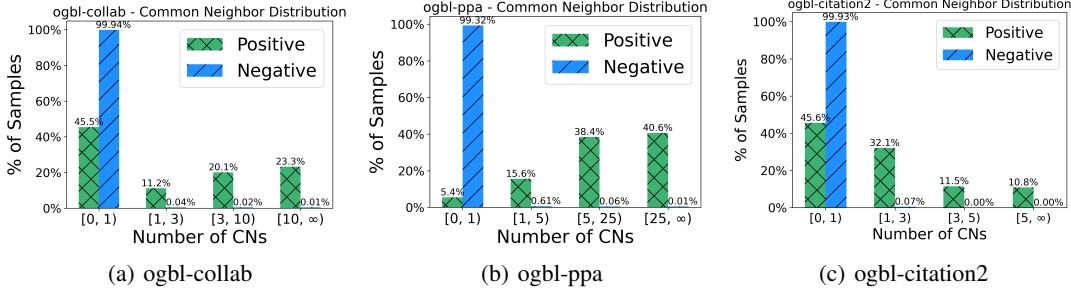

Figure 1: Common neighbor distribution for the positive and negative test samples for the ogbl-collab, ogbl-ppa, and ogbl-citation2 datasets under the existing evaluation setting.

1. **Lower than Actual Performance**. We observe that the current performance of multiple models is underreported. For some methods, such as standard GNNs, this is due to poor hyperparameter tuning. Once properly tuned, they can even achieve the best overall performance on some metrics (see SAGE [16] in Table 1). Furthermore, for other methods like Neo-GNN [14] we can achieve around an 8.5 point increase in Hits@50 on ogbl-collab relative to the originally reported performance. This results in Neo-GNN achieving the best overall performance on ogbl-collab in our study (see Table 2). Such problems obscure the true performance of different models, making it difficult to draw reliable conclusions from the current results.

2. **Lack of Unified Settings**. For Cora, Citeseer, and Pubmed datasets [17], there exists no unified data split and evaluation metrics used for each individually. For the data split, some works [18, 19] use a single fixed train/valid/test split with percentages 85/5/10%. More recent works [13, 15] use 10 random splits of size 70/10/20%. In terms of the evaluation metrics, some studies [13, 15] use ranking-based metrics such as MRR or Hits@K while others [20, 19] report the area under the curve (AUC). This is despite multiple studies that argue that AUC is a poor metric for evaluating link prediction [21, 22]. Additionally, for both planetoid (i.e., Cora, Citeseer and Pubmed) and ogbl-collab dataset, some methods incorporate validation edges during the testing phase [13], whereas others [14], exclude them. This lack of a unified setting makes it difficult to compare those works and hampers our ability to determine which methods perform best on these datasets.

3. **Unrealistic Evaluation Setting**. During the evaluation, we are given a set of true samples (i.e., positive samples) and a set of false samples (i.e., negative samples). We are tasked with learning a classifier $f$ that assigns a higher probability to the positive samples than the negatives. The current evaluation setting uses the same set of randomly selected negative samples for each positive sample. We identify two potential problems with the current evaluation procedure. **(1)** It is not aligned with real-world settings. In a real-world scenario, we typically care about predicting links for a specific node. For example, in friend recommendations, we aim to recommend friends for a specific user $u$. To evaluate such models for $u$, we strive to rank node pairs including $u$. However, this does not hold in the current setting as $u$ is not included in most of the negative samples. **(2)** The current evaluation setting makes the task too easy. As such, it may not reflect the model performance in real-world applications. This is because the nodes in a randomly selected negative "node pair" are likely to be unrelated to each other. As shown in Figure 1, almost all negative samples in the test data have no common neighbors, a typically strong heuristic, making them trivial to classify them.

To account for these issues, we propose to first conduct a fair and reproducible evaluation among current link prediction methods under the existing evaluation setting. We then design a new evaluation strategy that is more aligned with a real-world setting and detail our results. Our key contributions are summarized below:

- **Reproducible and Fair Comparison**. We conduct a fair comparison of different models across multiple common datasets. To ensure a fair comparison, we tune all models on the same set of hyperparameters. We further evaluate different models using multiple types of evaluation

metrics. For the Planetoid datasets [17], we further use a unified data split to facilitate a point of comparison between models. To the best of our knowledge, there are no recent efforts to comprehensively benchmark link prediction methods (several exist for KGC [10, 9, 8]). Furthermore, we open-source the implementation in our analysis to enable others in their analyses.

- **New Evaluation Setting**. We recognize that the current negative sampling strategy used in evaluation is unrealistic and easy. To counter these issues, we first use a more realistic setting of tailoring the negatives to each positive sample. This is achieved by restricting them to be corruptions of the positive sample (i.e., containing one of its two nodes as defined in Eq. (3)). Given the prohibitive cost of utilizing all possible corruptions, we opt instead to only rank against $K$ negatives for each positive sample. In order to choose the most relevant and difficult corruptions, we propose a **He**uristic **R**elated Sampling **T**echnique (HeaRT), which selects them based on a combination of multiple heuristics. This creates a more challenging task than the previous evaluation strategy and allows us to better assess the capabilities of current methods.

The rest of the paper is structured as follows. In Section 2 we introduce the models, datasets, and settings used for conducting a fair comparison between methods. In Section 3 we show the results of the fair comparison under the existing evaluation setting and discuss our main observations. Lastly, in Section 4 we introduce our new evaluation setting. We then detail and discuss the performance of different methods using our new setting.

## 2 Preliminaries

### 2.1 Task Formulation

In this section we formally define the task of link prediction. Let $\mathcal{G} = \{\mathcal{V}, \mathcal{E}\}$ be a graph where $\mathcal{V}$ and $\mathcal{E}$ are the set of nodes and edges in the graph, respectively. Furthermore, let $X \in \mathcal{R}^{|V| \times d}$ be a set of $d$-dimensional features for each node. Link prediction aims to predict the likelihood of a link existing between two nodes given the structural and feature information. For a pair of nodes $u$ and $v$, the probability of a link existing, $p(u, v)$, is therefore given by:

$$p(u, v) = p(u, v \mid \mathcal{G}, X). \tag{1}$$

Traditionally, $p(u, v)$ was estimated via non-learnable heuristic methods [23, 24]. More recently, methods that use learnable parameters have gained popularity [12, 13]. These methods attempt to estimate $p(u, v)$ via a learnable function $f$ such that:

$$p(u, v) = f(u, v \mid \mathcal{G}, X, \Theta), \tag{2}$$

where $\Theta$ represents a set of learnable parameters. A common choice of $f$ are graph neural networks [25]. In the next subsection we detail the various link prediction methods used in this study.

### 2.2 Link Prediction Methods

In this section we given an overview of the different methods used in this study. Conventional methods [23, 24] often exploit hand-craft graph structural properties (i.e., heuristics) between node pairs. GNNs attempt to learn the structural information to facilitate link prediction [26, 15, 13]. Given the strong performance of pairwise-based heuristics [14, 15], some recent works use both GNNs and pairwise information, demonstrating strong performance.

For our study, we consider both traditional and state-of-the-art GNN-based models. They can be roughly organized into four categories. **1) Heuristic methods**: Common Neighbor (CN) [27], Adamic Adar (AA) [28], Resource Allocation (RA) [29], Shortest Path [24], and Katz [30]. These methods define a score to indicate the link existence based on the graph structure. Among them, CN, AA, and RA are based on the common neighbors, while Shortest Path and Katz are based on the path information. **2) Embedding methods**: Matrix factorization (MF) [23], Multilayer Perceptron (MLP) and Node2Vec [31]. These methods are trained to learn low-dimensional node embeddings that are used to predict the likelihood of node pairs existing. **3) GNN methods**: GCN [32], GAT [18], SAGE [16], and GAE [20]. These methods attempt to integrate the multi-hop graph structure based on the message passing paradigm. **4) GNN + Pairwise Information methods**: Standard GNN methods, while powerful, are not able to capture link-specific information [26]. As such,

works have been proposed that augment GNN methods by including additional information to better capture the relation between the nodes in the link we are predicting. SEAL [26], BUDDY [13], and NBFNet [19] use the subgraph features. Neo-GNN [14], NCN [15], and NCNC [15] are based on common neighbor information. Lastly, PEG [33] uses the positional encoding derived from the graph structure.

### 2.3 Datasets and Experimental Settings

In this section we summarize the datasets and evaluation and training settings. We note that the settings depend on the specific dataset. More details are given in Appendix C.

**Datasets**. We limit our experiments to homogeneous graphs, which are the most commonly used datasets for link prediction. This includes the small-scale datasets, i.e., Cora, Citeseer, Pubmed [17], and large-scale datasets in the OGB benchmark [34], i.e., ogbl-collab, ogbl-ddi, ogbl-ppa, and ogbl-citation2. We summarize the statistics and split ratio of each dataset in Appendix C.

**Metrics**. For evaluation, we use both the area under the curve (AUC) and ranking-based metrics, i.e., mean reciprocal rank (MRR) and Hits@K. For Cora, Citeseer, and Pubmed we adopt $K \in \{1, 3, 10, 100\}$. We note that $K = 100$ is reported in some recent works [13, 15]). However due to the small number of negatives used during evaluation (e.g., $\approx 500$ for Cora and Citeseer) $K = 100$ is likely not informative. For the OGB datasets, we adopt $K \in \{20, 50, 100\}$ to keep consistent with the original study [34]. Please see Appendix B.1 for the formal definitions of the various evaluation metrics.

**Hyperparameter Ranges**. We conduct a grid hyperparameter search across a comprehensive range of values. For Cora, Citeseer, and Pubmed this includes: learning rate (0.01, 0.001), dropout (0.1, 0.3, 0.5), weight decay (1e-4, 1e-7, 0), number of model layers (1, 2, 3), number of prediction layers (1, 2, 3), and the embedding size (128, 256). Due to the large size of the OGB datasets, it's infeasible to tune over such a large range. Therefore, following the most commonly used settings among published hyperparameters, we fix the weight decay to 0, the number of model and prediction layers to be 3, and the embedding size to be 256. The best hyperparameters are chosen based on the validation performance. We note that several exceptions exist to these ranges when they result in significant performance degradations (see Appendix C for more details). We further follow the existing setting and only sample one negative sample per positive sample during training.

**Existing Evaluation Settings**. In the evaluation stage, the same set of randomly sampled negatives are used for all positive samples. We note that one exception is ogbl-citation2, where they randomly sample 1000 negative samples per positive sample. For Cora, Citeseer, and Pubmed the number of negative samples is equal to the number of positive samples. For the OGB datasets, we use the existing fixed set of randomly chosen negatives found in [34]. Furthermore, for ogbl-collab we follow the existing protocol [34] and include the validation edges in the training graph during testing. This setting is adopted on ogbl-collab under both the existing and new evaluation setting.

## 3 Fair Comparison Under the Existing Setting

In this section, we conduct a fair comparison among link prediction methods. This comparison is spurred by the multiple pitfalls noted in Section 1, which include lower-than-actual model performance, multiple data splits, and inconsistent evaluation metrics. These pitfalls hinder our ability to fairly compare different methods. To rectify this, we conduct a fair comparison adhering to the settings listed in section 2.3.

The results are split into two tables. The results for Cora, Citeseer, and Pubmed are shown in Table 1 and OGB in Table 2. For simplicity, we only present the AUC and MRR for Cora, Citeseer, and Pubmed. For OGB datasets, we include AUC and the original ranking metric reported in [34] to allow a convenient comparison (Hits@20 for ogbl-ddi, Hits@50 for ogbl-collab, Hits@100 for ogbl-ppa, and MRR for ogbl-citation2). We use ">24h" to denote methods that require more than 24 hours for either training one epoch or evaluation. OOM indicates that the algorithm requires over 50Gb of GPU memory. Since ogbl-ddi has no node features, we mark the MLP results with a "N/A". Additional results in terms of other metrics are presented in Appendix F. We have several noteworthy observations concerning the methods, the datasets, the evaluation settings, and the overall results. We highlight the main observations below.

Table 1: Results on Cora, Citeseer, and Pubmed(%) under the existing evaluation setting. Highlighted are the results ranked **first**, **second**, and **third**.

| | Models | Cora MRR | Cora AUC | Citeseer MRR | Citeseer AUC | Pubmed MRR | Pubmed AUC |
|---|---|---|---|---|---|---|---|
| Heuristic | CN | 20.99 | 70.85 | 28.34 | 67.49 | 14.02 | 63.9 |
| | AA | 31.87 | 70.96 | 29.37 | 67.49 | 16.66 | 63.9 |
| | RA | 30.79 | 70.96 | 27.61 | 67.48 | 15.63 | 63.9 |
| | Shortest Path | 12.45 | 81.08 | 31.82 | 75.5 | 7.15 | 74.64 |
| | Katz | 27.4 | 81.17 | 38.16 | 75.37 | 21.44 | 74.86 |
| Embedding | Node2Vec | 37.29 ± 8.82 | 90.97 ± 0.64 | 44.33 ± 8.99 | 94.46 ± 0.59 | 34.61 ± 2.48 | 93.14 ± 0.18 |
| | MF | 14.29 ± 5.79 | 80.29 ± 2.26 | 24.80 ± 4.71 | 75.92 ± 3.25 | 19.29 ± 6.29 | 93.06 ± 0.43 |
| | MLP | 31.21 ± 7.90 | 95.32 ± 0.37 | 43.53 ± 7.26 | 94.45 ± 0.32 | 16.52 ± 4.14 | 98.34 ± 0.10 |
| GNN | GCN | 32.50 ± 6.87 | 95.01 ± 0.32 | 50.01 ± 6.04 | 95.89 ± 0.26 | 19.94 ± 4.24 | 98.69 ± 0.06 |
| | GAT | 31.86 ± 6.08 | 93.90 ± 0.32 | 48.69 ± 7.53 | 96.25 ± 0.20 | 18.63 ± 7.75 | 98.20 ± 0.07 |
| | SAGE | 37.83 ± 7.75 | 95.63 ± 0.27 | 47.84 ± 6.39 | 97.39 ± 0.15 | 22.74 ± 5.47 | 98.87 ± 0.04 |
| | GAE | 29.98 ± 3.21 | 95.08 ± 0.33 | 63.33 ± 3.14 | 97.06 ± 0.22 | 16.67 ± 0.19 | 97.47 ± 0.08 |
| GNN+Pairwise Info | SEAL | 26.69 ± 5.89 | 90.59 ± 0.75 | 39.36 ± 4.99 | 88.52 ± 1.40 | 38.06 ± 5.18 | 97.77 ± 0.40 |
| | BUDDY | 26.40 ± 4.40 | 95.06 ± 0.36 | 59.48 ± 8.96 | 96.72 ± 0.26 | 23.98 ± 5.11 | 98.2 ± 0.05 |
| | Neo-GNN | 22.65 ± 2.60 | 93.73 ± 0.36 | 53.97 ± 5.88 | 94.89 ± 0.60 | 31.45 ± 3.17 | 98.71 ± 0.05 |
| | NCN | 32.93 ± 3.80 | 96.76 ± 0.18 | 54.97 ± 6.03 | 97.04 ± 0.26 | 35.65 ± 4.60 | 98.98 ± 0.04 |
| | NCNC | 29.01 ± 3.83 | 96.90 ± 0.28 | 64.03 ± 3.67 | 97.65 ± 0.30 | 25.70 ± 4.48 | 99.14 ± 0.03 |
| | NBFNet | 37.69 ± 3.97 | 92.85 ± 0.17 | 38.17 ± 3.06 | 91.06 ± 0.15 | 44.73 ± 2.12 | 98.34 ± 0.02 |
| | PEG | 22.76 ± 1.84 | 94.46 ± 0.34 | 56.12 ± 6.62 | 96.15 ± 0.41 | 21.05 ± 2.85 | 96.97 ± 0.39 |

Table 2: Results on OGB datasets (%) under the existing evaluation setting. Highlighted are the results ranked **first**, **second**, and **third**.

| | Models | ogbl-collab Hits@50 | ogbl-collab AUC | ogbl-ddi Hits@20 | ogbl-ddi AUC | ogbl-ppa Hits@100 | ogbl-ppa AUC | ogbl-citation2 MRR |
|---|---|---|---|---|---|---|---|---|
| Heuristic | CN | 61.37 | 82.78 | 17.73 | 95.2 | 27.65 | 97.22 | 74.3 |
| | AA | 64.17 | 82.78 | 18.61 | 95.43 | 32.45 | 97.23 | 75.96 |
| | RA | 63.81 | 82.78 | 6.23 | 96.51 | 49.33 | 97.24 | 76.04 |
| | Shortest Path | 46.49 | 96.51 | 0 | 59.07 | 0 | 99.13 | >24h |
| | Katz | 64.33 | 90.54 | 17.73 | 95.2 | 27.65 | 97.22 | 74.3 |
| Embedding | Node2Vec | 49.06 ± 1.04 | 96.24 ± 0.15 | 34.69 ± 2.90 | 99.78 ± 0.04 | 26.24 ± 0.96 | 99.77 ± 0.00 | 45.04 ± 0.10 |
| | MF | 41.81 ± 1.67 | 83.75 ± 1.77 | 23.50 ± 5.35 | 99.46 ± 0.10 | 28.4 ± 4.62 | 99.46 ± 0.10 | 50.57 ± 12.14 |
| | MLP | 35.81 ± 1.08 | 95.91 ± 0.08 | N/A | N/A | 0.45 ± 0.04 | 90.23 ± 0.00 | 38.07 ± 0.09 |
| GNN | GCN | 54.96 ± 3.18 | 97.89 ± 0.06 | 49.90 ± 7.23 | 99.86 ± 0.03 | 29.57 ± 2.90 | 99.84 ± 0.03 | 84.85 ± 0.07 |
| | GAT | 55.00 ± 3.28 | 97.11 ± 0.09 | 31.88 ± 8.83 | 99.63 ± 0.21 | OOM | OOM | OOM |
| | SAGE | 59.44 ± 1.37 | 98.08 ± 0.03 | 49.84 ± 15.56 | 99.96 ± 0.00 | 41.02 ± 1.94 | 99.82 ± 0.00 | 83.06 ± 0.09 |
| | GAE | OOM | OOM | 7.09 ± 6.02 | 75.34 ±15.96 | OOM | OOM | OOM |
| GNN+Pairwise Info | SEAL | 63.37 ± 0.69 | 95.65 ± 0.29 | 25.25 ± 3.90 | 97.97 ± 0.19 | 48.80 ± 5.61 | 99.79 ± 0.02 | 86.93 ± 0.43 |
| | BUDDY | 64.59 ± 0.46 | 96.52 ± 0.40 | 29.60 ± 4.75 | 99.81 ± 0.02 | 47.33 ± 1.96 | 99.56 ± 0.02 | 87.86 ± 0.18 |
| | Neo-GNN | 66.13 ± 0.61 | 98.23 ± 0.05 | 20.95 ± 6.03 | 98.06 ± 2.00 | 48.45 ± 1.01 | 97.30 ± 0.14 | 83.54 ± 0.32 |
| | NCN | 63.86 ± 0.51 | 97.83 ± 0.04 | 76.52 ± 10.47 | 99.97 ± 0.00 | 62.63 ± 1.15 | 99.95 ± 0.01 | 89.27 ± 0.05 |
| | NCNC | 65.97 ± 1.03 | 98.20 ± 0.05 | 70.23 ± 12.11 | 99.97 ± 0.01 | 62.61 ± 0.76 | 99.97 ± 0.01 | 89.82 ± 0.43 |
| | NBFNet | OOM | OOM | >24h | >24h | OOM | OOM | OOM |
| | PEG | 49.02 ± 2.99 | 94.45 ± 0.89 | 30.28 ± 4.92 | 99.45 ± 0.04 | OOM | OOM | OOM |

**Observation 1: Better than Reported Performance.** We find that for some models we are able to achieve superior performance compared to what is reported by recent studies. For instance, in our study Neo-GNN [14] achieves the best overall test performance on ogbl-collab with a Hits@50 of 66.13. In contrast, the reported performance in [14] is only 57.52, which would rank seventh under our current setting. This is because the original study [14] does not follow the standard setting of including validation edges in the graph during testing. This setting, as noted in Section 2.3, is used by all other methods on ogbl-collab. However it was omitted by [14], resulting in lower reported performance. Furthermore, on ogbl-citation2 [34], our results for the heuristic methods are typically around 75% MRR. This significantly outperforms previously reported results, which report an MRR of around 50% [26, 13]. The disparity arises as previous studies treat the ogbl-citation2 as a directed graph when applying heuristic methods. However, for GNN-based methods, ogbl-citation2 is typically converted to a undirected graph. We remedy this by also converting ogbl-citation2 to an undirected graph when computing the heuristics, leading to a large increase in performance.

Furthermore, with proper tuning, conventional baselines like GCN [25] and GAE [20] generally exhibit enhanced performance relative to what was originally reported across all datasets. For example, we find that GAE can achieve the second best MRR on Citeseer and GCN the third best Hits@20 on ogbl-ddi. A comparison of the reported results and ours are shown in Table 3. We note that we report AUC for Cora, Citeseer, Pubmed as it was used in the original study. These

Table 3: Comparison of ours and the reported results for GCN and GAE.

| GCN | ogbl-collab
Hits@50 | ogbl-ppa
Hits@100 | ogbl-ddi
Hits@20 | ogbl-citation2
MRR | GAE | Cora
AUC | Citeseer
AUC | Pubmed
AUC |
|---|---|---|---|---|---|---|---|---|
| Reported | 47.14 ± 1.45 | 18.67 ± 1.32 | 37.07 ± 5.07 | 84.74 ± 0.21 | Reported | 91.00 ± 0.01 | 89.5 ± 0.05 | 96.4 ± 0.00 |
| Ours | **54.96 ± 3.18** | **29.57 ± 2.90** | **49.90 ± 7.23** | **84.85 ± 0.07** | Ours | **95.08 ± 0.33** | **97.06 ± 0.22** | **97.47 ± 0.08** |

observations suggest that the performance of various methods are better than what was reported in their initial publications. However, many studies [13, 15, 26] only report the original performance for comparison, which has the potential to lead to inaccurate conclusions.

**Observation 2: Divergence from Reported Results on ogbl-ddi.** We observe that our results in Table 2 for ogbl-ddi differ from the reported results. Outside of GCN, which reports better performance, most other GNN-based methods report a lower-than-reported performance. For example, for BUDDY we only achieve a Hits@20 of 29.60 vs. the reported 78.51 (see Appendix D for a comprehensive comparison among methods). We find that the reason for this difference depends on the method. BUDDY [13] reported [2] using 6 negatives per positive sample during training, leading to an increase in performance. Neo-GNN [14] first pretrains the GNN under the link prediction task, and then uses the pretrained model as the initialization for Neo-GNN.[3] For a fair comparison among methods, we only use 1 negative per positive sample in training and we don't apply the pretraining. For other methods, we find that a weak relationship between the validation and test performance complicates the tuning process, making it difficult to find the optimal hyperparameters. Please see Appendix E for a more in-depth study and discussion.

**Observation 3: High Model Standard Deviation.** The results in Tables 1 and 2 present the mean performance and standard deviation when training over 10 seeds. Generally, we find that for multiple datasets the standard deviation of the ranking metrics is often high for most models. For example, the standard deviation for MRR can be as high as 8.82, 8.96, or 7.75 for Cora, Citeseer, and Pubmed, respectively. Furthermore, on ogbl-ddi the standard deviation of Hits@20 reaches as high as 10.47 and 15.56. A high variance indicates unstable model performance. This makes it difficult to compare results between methods as the true performance lies in a larger range. This further complicates replicating model performance, as even large differences with the reported results may still fall within variance (see observation 2). Later in Section 4.3 we find that our new evaluation can reduce the model variance for all datasets (see Table 6). This suggests that the high variance is related to the current evaluation procedure.

**Observation 4: Inconsistency of AUC vs. Ranking-Based Metrics.** The AUC score is widely adopted to evaluate recent advanced link prediction methods [20, 19]. However, from our results in Tables 1 and 2 we observe that there exists a disparity between AUC and ranking-based metrics. In some cases, the AUC score can be high when the ranking metric is very low or even 0. For example, the Shortest Path heuristic records a Hits@K of 0 on ogbl-ppa. However, the AUC score is $> 99\%$. Furthermore, even though RA records the third and fifth best performance on ogbl-ppa and ogbl-collab, respectively, it has a lower AUC score than Shortest Path on both. Previous works [22, 21] argued that AUC is not a proper metric for link prediction. This is due to the inapplicability of AUC for highly imbalanced problems [35, 36].

## 4 New Evaluation Setting

In this section, we introduce a new setting for evaluating link prediction methods. We first discuss the unrealistic nature of the current evaluation setting in Section 4.1. Based on this, we present our new evaluation setting in Section 4.2, which aims to align better with real-world scenarios. Lastly, in Section 4.3, we present and discuss the results based on our new evaluation setting.

### 4.1 Issues with the Existing Evaluation Setting

The existing evaluation procedure for link prediction is to rank a positive sample against a set of $K$ randomly selected negative samples. The same set of $K$ negatives are used for all positive samples

---

[2]https://github.com/melifluos/subgraph-sketching
[3]https://github.com/seongjunyun/Neo-GNNs

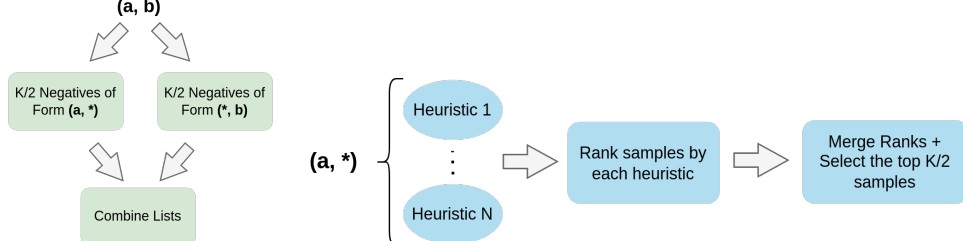

(a) Negative sample genera-
tion for one positive sample.

(b) Process of determining negative samples that contain a node $a$.

Figure 2: Pipeline for generating the hard negative samples for a positive sample (a, b).

(with the exception of ogbl-citation2 which uses 1000 per positive sample). We demonstrate that there are multiple issues with this setting, making it difficult to properly evaluate the effectiveness of current models.

**Issue 1: Non-Personalized Negative Samples.** The existing evaluation setting uses the same set of negative samples for all positive samples (outside of ogbl-citation2). This strategy, referred to as global negative sampling [37], is not a commonly sought objective. Rather, we are often more interested in predicting links that will occur for a specific node. Take, for example, a social network that connects users who are friends. In this scenario, we may be interested in recommending new friends to a user $u$. This requires learning a classifier $f$ that assigns a probability to a link existing. When evaluating this task, we want to rank links where $u$ connects to an existing friend above those where they don't. For example, if $u$ is friends with $a$ but not $b$, we hope that $f(u, a) > f(u, b)$. However, the existing evaluation setting doesn't explicitly test for this. Rather it compares a true sample $(u, a)$ with a potentially unrelated negative sample, e.g., $(c, d)$. This is not aligned with the real-world usage of link prediction on such graphs.

**Issue 2: Easy Negative Samples.** The existing evaluation setting randomly selects negative samples to use. However given the large size of most graphs (see Table 7 in Appendix C), randomly sampled negatives are likely to choose two nodes that bear no relationship to each other. Such node pairs are trivial to classify. We demonstrate this by plotting the distribution of common neighbors (CN), a strong heuristic, for all positive and negative test samples in Figure 1. Almost all the negative samples contain no CNs, making them easy to classify. We further show that the same problem afflicts even the smaller datasets in Figure 3 in Appendix A.

These observations suggest that a more realistic evaluation strategy is desired. At the core of this challenge is which negative samples to use during evaluation. We discuss our design for solving this in the next subsection.

## 4.2 Heuristic Related Sampling Technique (HeaRT)

In this subsection, we introduce new strategy for evaluating link prediction methods. To address the concerns outlined in Section 4.1, we design a new method for sampling negatives during evaluation. Our strategy, HeaRT, solves these challenges by: (a) personalizing the negatives to each sample and (b) using heuristics to select hard negative samples. This allows for the negative samples to be directly related to each positive sample while also being non-trivial. We further discuss how to ensure that the negative samples are both *personalized* and *non-trivial* for a specific positive sample.

From our discussion in Section 4.1, we are motivated in personalizing the negatives to each positive sample. Since the positive samples in the current datasets are node pairs, we seek to personalize the negatives to both nodes in the positive sample. Extending our example in Section 4.1, this is analogous to restricting the negatives to contain one of the two users from the original friendship pair. As such, for a positive sample $(u, a)$, the negative samples will belong to the set:

$$S(u, a) = \{(u', a) \mid u' \in \mathcal{V}\} \cup \{(u, a') \mid a' \in \mathcal{V}\}, \tag{3}$$

where $\mathcal{V}$ is the set of nodes. This is similar to the setting used for knowledge graph completion (KGC) [38] which uses all such samples for evaluation. However, one drawback of evaluating each

Table 4: Results on Cora, Citeseer, and Pubmed (%) under HeaRT. Highlighted are the results ranked **first**, **second**, and **third**.

| | Models | Cora | | Citeseer | | Pubmed | |
|---|---|---|---|---|---|---|---|
| | | MRR | Hits@10 | MRR | Hits@10 | MRR | Hits@10 |
| Heuristic | CN | 9.78 | 20.11 | 8.42 | 18.68 | 2.28 | 4.78 |
| | AA | 11.91 | 24.10 | 10.82 | 22.20 | 2.63 | 5.51 |
| | RA | 11.81 | 24.48 | 10.84 | 22.86 | 2.47 | 4.9 |
| | Shortest Path | 5.04 | 15.37 | 5.83 | 16.26 | 0.86 | 0.38 |
| | Katz | 11.41 | 22.77 | 11.19 | 24.84 | 3.01 | 5.98 |
| Embedding | Node2Vec | 14.47 ± 0.60 | 32.77 ± 1.29 | 21.17 ± 1.01 | 45.82 ± 2.01 | 3.94 ± 0.24 | 8.51 ± 0.77 |
| | MF | 6.20 ± 1.42 | 15.26 ± 3.39 | 7.80 ± 0.79 | 16.72 ± 1.99 | 4.46 ± 0.32 | 9.42 ± 0.87 |
| | MLP | 13.52 ± 0.65 | 31.01 ± 1.71 | 22.62 ± 0.55 | 48.02 ± 1.79 | 6.41 ± 0.25 | 15.04 ± 0.67 |
| GNN | GCN | **16.61 ± 0.30** | **36.26 ± 1.14** | 21.09 ± 0.88 | 47.23 ± 1.88 | 7.13 ± 0.27 | 15.22 ± 0.57 |
| | GAT | 13.84 ± 0.68 | 32.89 ± 1.27 | 19.58 ± 0.84 | 45.30 ± 1.3 | 4.95 ± 0.14 | 9.99 ± 0.64 |
| | SAGE | 14.74 ± 0.69 | 34.65 ± 1.47 | 21.09 ± 1.15 | 48.75 ± 1.85 | **9.40 ± 0.70** | **20.54 ± 1.40** |
| | GAE | **18.32 ± 0.41** | **37.95 ± 1.24** | **25.25 ± 0.82** | **49.65 ± 1.48** | 5.27 ± 0.25 | 10.50 ± 0.46 |
| GNN+Pairwise Info | SEAL | 10.67 ± 3.46 | 24.27 ± 6.74 | 13.16 ± 1.66 | 27.37 ± 3.20 | 5.88 ± 0.53 | 12.47 ± 1.23 |
| | BUDDY | 13.71 ± 0.59 | 30.40 ± 1.18 | 22.84 ± 0.36 | 48.35 ± 1.18 | 7.56 ± 0.18 | 16.78 ± 0.53 |
| | Neo-GNN | 13.95 ± 0.39 | 31.27 ± 0.72 | 17.34 ± 0.84 | 41.74 ± 1.18 | **7.74 ± 0.30** | **17.88 ± 0.71** |
| | NCN | 14.66 ± 0.95 | 35.14 ± 1.04 | **28.65 ± 1.21** | **53.41 ± 1.46** | 5.84 ± 0.22 | 13.22 ± 0.56 |
| | NCNC | 14.98 ± 1.00 | **36.70 ± 1.57** | **24.10 ± 0.65** | **53.72 ± 0.97** | **8.58 ± 0.59** | **18.81 ± 1.16** |
| | NBFNet | 13.56 ± 0.58 | 31.12 ± 0.75 | 14.29 ± 0.80 | 31.39 ± 1.34 | >24h | >24h |
| | PEG | **15.73 ± 0.39** | 36.03 ± 0.75 | 21.01 ± 0.77 | 45.56 ± 1.38 | 4.4 ± 0.41 | 8.70 ± 1.26 |

positive sample against the entire set of possible corruptions is the high computational cost. To mitigate this issue we consider only utilizing a small subset of $S(u, a)$ during evaluation.

*The key challenge is how to generate a subset of $S(u, a)$.* If we randomly sample from $S(u, a)$, we risk only utilizing easy negative samples. This is one of the issues of the existing evaluation setting (see Issue 2 in Section 4.1), whereby randomly selecting negatives, they unknowingly produce negative samples that are too easy. We address this by selecting the negative samples via a combination of multiple heuristics. Since heuristics typically correlate well with performance, we ensure that the negative samples will be non-trivial to classify. This is similar to the concept of candidate generation [39, 40], which only ranks a subset of candidates that are most likely to be true.

An overview of the generation process is given in Figure 2. For each positive sample, we generate $K$ negative samples. To allow personalization to both nodes in the positive sample equally, we sample $K/2$ negatives with each node. For the heuristics, we consider RA [29], PPR [41], and feature similarity. A more detailed discussion on the negative sample generation is given in Appendix G. It's important to note that our work centers specifically on negative sampling during the evaluation stage (validation and test). This is distinct from prior work that concerns the negatives sampled used during the training phase [42, 43]. As such, the training process remains unaffected under both the existing and HeaRT setting.

## 4.3 Results and Discussion

In this subsection we present our results when utilizing HeaRT. We follow the parameter ranges introduced in Section 2.3. For all datasets we use $K = 500$ negative samples per positive sample during evaluation. Furthermore for ogbl-ppa we only use a small subset of the validation and test positive samples (100K each) for evaluation. This is because the large size of the validation and test sets (see Table 7 in Appendix C) makes HeaRT prohibitively expensive.

The results are shown in Table 4 (Cora, Citeseer, Pubmed) and Table 5 (OGB). For simplicity, we only include the MRR and Hits@10 for Cora, Citeseer, Pubmed, and the MRR and Hits@20 for OGB. Additional results for other metrics can be found in Appendix I. We note that most datasets, outside of ogbl-ppa, exhibit much lower performance than under the existing setting. This is though we typically use much fewer negative samples in the new setting, implying that the negative samples produced by HeaRT are much harder. We highlight the main observations below.

**Observation 1: Better Performance of Simple Models**. We find that under HeaRT, "simple" baseline models (i.e., heuristic, embedding, and GNN methods) show a greater propensity to outperform their counterparts via ranking metrics than under the existing setting. Specifically, we focus on MRR in Table 1, 4, and 5, and the corresponding ranking-based metrics in Table 2. Under the existing

Table 5: Results on OGB datasets (%) under HeaRT. Highlighted are the results ranked **first**, **second**, and **third**.

| Models | ogbl-collab | | ogbl-ddi | | ogbl-ppa | | ogbl-citation2 | |
|---|---|---|---|---|---|---|---|---|
| | MRR | Hits@20 | MRR | Hits@20 | MRR | Hits@20 | MRR | Hits@20 |
| CN | 12.60 | 27.51 | 6.71 | 38.69 | 25.70 | 68.25 | 17.11 | 41.73 |
| AA | 16.40 | 32.65 | 6.97 | 39.75 | 26.85 | 70.22 | 17.83 | 43.12 |
| RA | 28.14 | 41.16 | 8.70 | 44.01 | 28.34 | 71.50 | 17.79 | 43.34 |
| Shortest Path | 46.71 | 46.56 | 0 | 0 | 0.54 | 1.31 | >24h | >24h |
| Katz | 47.15 | 48.66 | 6.71 | 38.69 | 25.70 | 68.25 | 14.10 | 35.55 |
| Node2Vec | 12.10 ± 0.20 | 25.85 ± 0.21 | 11.14 ± 0.95 | 63.63 ± 2.05 | 18.33 ± 0.10 | 53.42 ± 0.11 | 14.67 ± 0.18 | 42.68 ± 0.20 |
| MF | 26.86 ± 1.74 | 38.44 ± 0.07 | 13.99 ± 0.47 | 59.50 ± 1.68 | 22.47 ± 1.53 | 70.71 ± 4.82 | 8.72 ± 2.60 | 29.64 ± 7.30 |
| MLP | 12.61 ± 0.66 | 23.05 ± 0.89 | N/A | N/A | 0.98 ± 0.00 | 1.47 ± 0.00 | 16.32 ± 0.07 | 43.15 ± 0.10 |
| GCN | 18.28 ± 0.84 | 32.90 ± 0.66 | 13.46 ± 0.34 | 64.76 ± 1.45 | 26.94 ± 0.48 | 68.38 ± 0.73 | 19.98 ± 0.35 | 51.72 ± 0.46 |
| GAT | 10.97 ± 1.16 | 29.58 ± 2.42 | 12.92 ± 0.39 | 66.83 ± 2.23 | OOM | OOM | OOM | OOM |
| SAGE | 20.89 ± 1.06 | 33.83 ± 0.93 | 12.60 ± 0.72 | 67.19 ± 1.18 | 27.27 ± 0.30 | 69.49 ± 0.43 | 22.05 ± 0.12 | 53.13 ± 0.15 |
| GAE | OOM | OOM | 3.49 ± 1.73 | 17.81 ± 9.80 | OOM | OOM | OOM | OOM |
| SEAL | 22.53 ± 3.51 | 36.48 ± 2.55 | 9.99 ± 0.90 | 49.74 ± 2.39 | 29.71 ± 0.71 | 76.77 ± 0.94 | 20.60 ± 1.28 | 48.62 ± 1.93 |
| BUDDY | 32.42 ± 1.88 | 45.62 ± 0.52 | 12.43 ± 0.50 | 58.71 ± 1.63 | 27.70 ± 0.33 | 71.50 ± 0.68 | 19.17 ± 0.20 | 47.81 ± 0.37 |
| Neo-GNN | 21.90 ± 0.65 | 38.40 ± 0.29 | 10.86 ± 2.16 | 51.94 ± 10.33 | 21.68 ± 1.14 | 64.81 ± 2.26 | 16.12 ± 0.25 | 43.17 ± 0.53 |
| NCN | 17.51 ± 2.50 | 37.07 ± 2.97 | 12.86 ± 0.78 | 65.82 ± 2.66 | 35.06 ± 0.26 | 81.89 ± 0.31 | 23.35 ± 0.28 | 53.76 ± 0.20 |
| NCNC | 19.02 ± 5.32 | 35.67 ± 6.78 | >24h | >24h | 33.52 ± 0.26 | 82.24 ± 0.40 | 19.61 ± 0.54 | 51.69 ± 1.48 |
| NBFNet | OOM | OOM | >24h | >24h | OOM | OOM | OOM | OOM |
| PEG | 15.68 ± 1.10 | 29.74 ± 0.95 | 12.05 ± 1.14 | 50.12 ± 6.55 | OOM | OOM | OOM | OOM |

setting, such methods only rank in the top three for any dataset a total of **5** times. However, under HeaRT this occurs **10** times. Furthermore, under the existing setting only **1** "simple" method ranks best overall while under HeaRT there are **4**. This suggests that recent advanced methods may have benefited from the easier negative samples in the existing setting.

Another interesting observation is that on ogbl-collab, heuristic methods are able to outperform more complicated models by a large margin. Specifically, we find that Katz is the best ranked method, Shortest Path the second, and RA the fourth. Furthermore, the MRR gap between the second ranked method (Shortest Path) and the third (BUDDY) is very large at 14.29 points. We observe that this result is caused by the dynamic nature of the graph, where node pairs that are linked in the training data may also be present as positive samples in the test. We further expound on this observation in Appendix H.

**Observation 2: Lower Model Standard Deviation**. We observed earlier that, under the existing evaluation setting, the model variance across seeds was high (see observation 3 in Section 3). This complicates model comparison as the model performance is unreliable. Interestingly, we find that HeaRT is able to dramatically reduce the variance for all datasets. We demonstrate this by first calculating the mean standard deviation across all models on each individual dataset. This was done for both evaluation settings with the results compared. As demonstrated in Table 6, the mean standard deviation decreases for all datasets. This is especially true for Cora, Citeseer, and Pubmed, which each decrease by over 85%. Such a large decrease in standard deviation is noteworthy as it allows for a more trustworthy and reliable comparison between methods.

We posit that this observation is caused by a stronger alignment between the positive and negative samples under our new evaluation setting. Under the existing evaluation setting, the same set of negative samples is used for all positive samples. One consequence of this is that a single positive sample may bear little to no relationship to the negative samples (see Section 4.1 for more discussion). However, under our new evaluation setting, the negatives for a positive sample are a subset of the corruptions of that sample. This allows for a more natural comparison via ranking-based metrics as the samples are more related and can be more easily compared.

Table 6: Mean model standard deviation for the existing setting and HeaRT. We use Hits@20 for ogbl-ddi, Hits@50 for ogbl-collab, Hits@100 for ogbl-ppa, and MRR otherwise.

| Dataset | Existing | HeaRT | % Change |
|---|---|---|---|
| Cora | 5.19 | 0.79 | -85% |
| Citeseer | 5.94 | 0.88 | -85% |
| Pubmed | 4.14 | 0.35 | -92% |
| ogbl-collab | 1.49 | 0.96 | -36% |
| ogbl-ppa | 2.13 | 0.36 | -83% |
| ogbl-ddi | 6.77 | 3.49 | -48% |
| ogbl-citation2 | 1.39 | 0.59 | -58% |

**Observation 3: Lower Model Performance**. We observe that the majority of datasets exhibit a significantly reduced performance in comparison to the existing setting. For example, under the existing setting, models typically achieve a MRR of around 30, 50, and 30 on Cora, Citeseer, and Pubmed (Table 1), respectively. However, under HeaRT the MRR for those datasets is typically around 20, 25, and 10 (Table 4). Furthermore for ogbl-citation2, the MRR of the best performing model falls from a shade under 90 on the existing setting to slightly over 20 on HeaRT. Lastly, we note that the performance on ogbl-ppa actually increases. This is because we only utilize a small subset of the total test set when evaluating on HeaRT, nullifying any comparison between the two settings.

These outcomes are observed despite HeaRT using much fewer negative samples than the original setting. This suggests that the negative samples generated by HeaRT are substantially more challenging than those used in the existing setting. This underscores the need to develop more advanced methodologies that can tackle harder negatives samples like in HeaRT.

## 5   Conclusion

In this work we have revealed several pitfalls found in recent works on link prediction. To overcome these pitfalls, we first establish a benchmarking that facilitates a fair and consistent evaluation across a diverse set of models and datasets. By doing so, we are able to make several illuminating observations about the performance and characteristics of various models. Furthermore, based on several limitations we observed in the existing evaluation procedure, we introduce a more practical setting called HeaRT (Heuristic Related Sampling Technique). HeaRT incorporates a more real-world evaluation setting, resulting in a better comparison among methods. By introducing a more rigorous and realistic assessment, HeaRT could guide the field towards more effective models, thereby advancing the state of the art in link prediction.

## 6   Acknowledgements

This research is supported by the National Science Foundation (NSF) under grant numbers CNS 2246050, IIS1845081, IIS2212032, IIS2212144, IOS2107215, DUE 2234015, III-2212145, III-2153326, DRL2025244 and IOS2035472, the Army Research Office (ARO) under grant number W911NF-21-1-0198, the Home Depot, Cisco Systems Inc, Amazon Faculty Award, Johnson&Johnson, JP Morgan Faculty Award and SNAP.

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
