

Figure 3: Common neighbor distribution for the positive and negative test samples for the Cora, Citeseer, Pubmed, and ogbl-ddi under the existing evaluation setting.

# A   Common Neighbor Distribution

In Figure 1 we demonstrate the common neighbor (CN) distribution among positive and negative test samples for ogbl-collab, ogbl-ppa, and ogbl-citation2. These results demonstrate that a vast majority of negative samples have no CNs. Since CNs is a typically good heuristic, this makes it easy to identify most negative samples.

We further present the CN distribution of Cora, Citeseer, Pubmed, and ogbl-ddi in Figure 3. The CN distribution of Cora, Citeseer, and Pubmed are consistent with our previous observations on the OGB datasets in Figure 1. We note that ogbl-ddi exhibits a different distribution with other datasets. As compared to the other datasets, most of the negative samples in ogbl-ddi have common neighbors. This is likely because ogbl-ddi is considerably denser than the other graphs. As shown in Table 7, the average node degree in ogbl-ddi is $625.68$, significantly larger than the second largest dataset ogbl-ppa with $105.25$. Thus, despite the random sampling of negative samples, the high degree of node connectivity within the ogbl-ddi graph predisposes a significant likelihood for the occurrence of common neighbors.

We also present the CN distributions under the HeaRT setting. The plots for Cora, Citeseer, Pubmed are shown in Figure 4. The plots for the OGB datasets are shown in Figure 5. We observe that the CN distribution of HeaRT is more aligned with the positive samples. This allows for a fairer evaluation setting by not favoring models that use CN information.

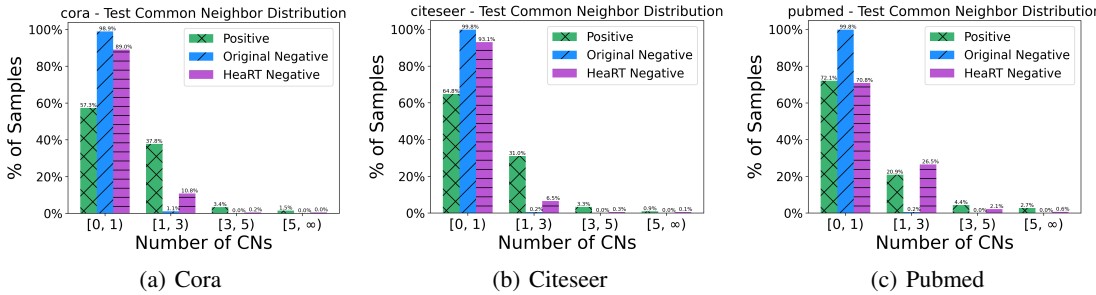

Figure 4: Common neighbor distribution for the positive negative samples under both evaluation settings for Cora, Citeseer, Pubmed.

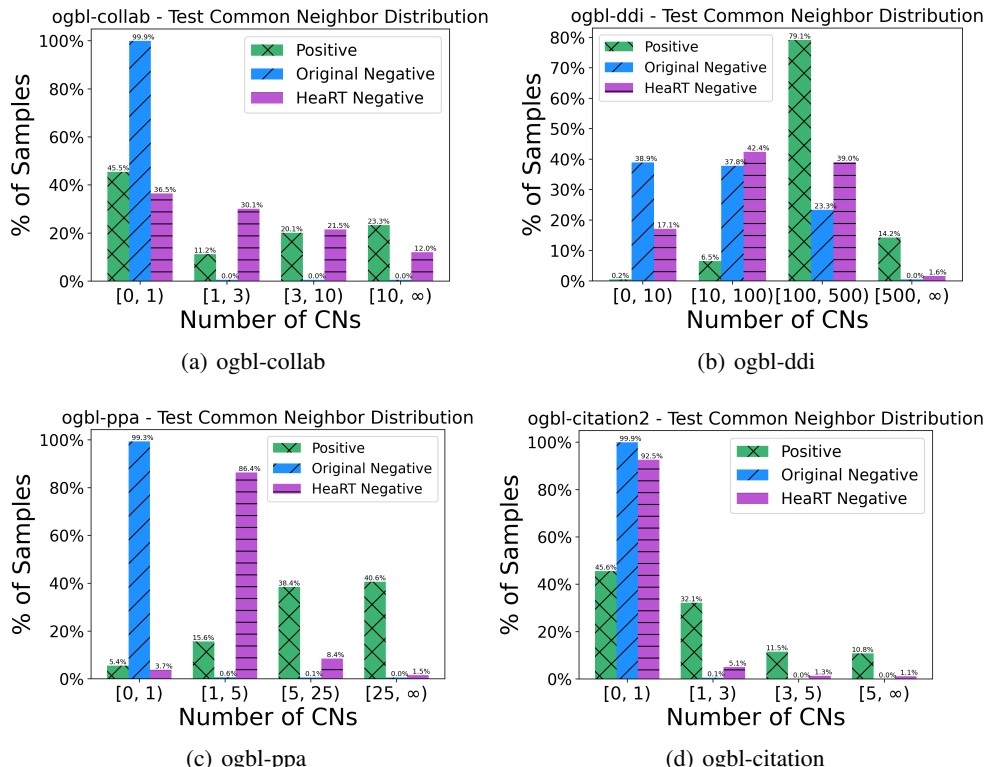

Figure 5: Common neighbor distribution for the positive negative samples under both evaluation settings for the OGB datasets.

# B Additional Definitions

## B.1 Evaluation Metrics

In this section we define the various evaluation metrics used. Given a single positive sample and $M$ negative samples, we first score each sample and then rank the positive sample among the negatives. The rank is then given by $\text{rank}_i$. I.e., a rank of 1 indicates that the positive sample has a higher score than all negatives. The hope is that the positive sample ranks above most or all negative samples. Various metrics make use of this rank. We use $N$ to denote the number of positive samples.

**Hits@K**. It measures whether the true positive is within the top K predictions or not: Hits@K $= \frac{1}{N} \sum_{i=1}^{N} \mathbf{1}(\text{rank}_i \leq \text{K})$. $\text{rank}_i$ is the rank of the $i$-th sample. The indicator function $\mathbf{1}$ is 1 if $\text{rank}_i \leq \text{K}$, and 0 otherwise.

**Mean Reciprocal Rank (MRR)**. It is the mean of the reciprocal rank over all positive samples: MRR $= \frac{1}{N} \sum_{i=1}^{N} \frac{1}{\text{rank}_i}$, where $\text{rank}_i$ is the rank of the $i$-th sample.

**AUC**. It measures the likelihood that a positive sample is ranked higher than a random negative sample: AUC $= \frac{\sum_{i \in \mathcal{D}^0} \sum_{j \in \mathcal{D}^1} \mathbf{1}(\text{rank}_i < \text{rank}_j)}{|\mathcal{D}^0| \cdot |\mathcal{D}^1|}$, where $\mathcal{D}^0$ is the set of positive samples, $\mathcal{D}^1$ is the set of negative samples, and $\text{rank}_i$ is the rank of the $i$-th sample. The indicator function $\mathbf{1}$ is 1 if $\text{rank}_i < \text{rank}_j$, and 0 otherwise.

### B.2 Negative Sampling

Since only positive links are observed, there is a need to generate negative links (i.e., edges that don't exist in $\mathcal{G}$) to both train and evaluate different models. We detail how these samples are generated in both training and evaluation.

**Training Negative Samples**. During training, the negative samples are randomly selected, with all nodes being equally likely to be selected. Let $\mathcal{V}$ and $\mathcal{E}$ be the set of nodes and edges in $\mathcal{G}$. Furthermore, we define $v \in \text{Rand}(\mathcal{V})$ as returning a random node in $\mathcal{V}$. A single negative sample $(a^-, b^-)$ is given by:

$$(a^-, b^-) = (\text{Rand}(\mathcal{V}), \text{Rand}(\mathcal{V})). \tag{4}$$

Typically one negative sample is generated per positive sample.

**Evaluation Negative Samples**. For the existing setting, a fixed set of randomly selected samples are used as negatives during evaluation. Furthermore, the same set of negative samples are used for each positive sample. This is equivalent to Eq. (4). The only exception is the ogbl-citation2 [34] dataset. For ogbl-citation2, each positive sample is only evaluated against its own set 1000 negative samples. For a positive sample, its negative samples are restricted to contain one of its two nodes (i.e., a corruption). The other node is randomly selected from $\mathcal{V}$. This is equivalent to selecting a set of random samples from the set $S(a, b)$ as defined in Eq. (3).

## C Datasets and Experimental Settings

### C.1 Datasets

Table 7: Statistics of datasets. The split ratio is for train/validation/test.

|  | Cora | Citeseer | Pubmed | ogbl-collab | ogbl-ddi | ogbl-ppa | ogbl-citation2 |
|---|---|---|---|---|---|---|---|
| #Nodes | 2,708 | 3,327 | 18,717 | 235,868 | 4,267 | 576,289 | 2,927,963 |
| #Edges | 5,278 | 4,676 | 44,327 | 1,285,465 | 1,334,889 | 30,326,273 | 30,561,187 |
| Mean Degree | 3.9 | 2.81 | 4.74 | 10.90 | 625.68 | 105.25 | 20.88 |
| Split Ratio | 85/5/10 | 85/5/10 | 85/5/10 | 92/4/4 | 80/10/10 | 70/20/10 | 98/1/1 |

The statistics of datasets are shown in Table 7. Generally, Cora, Citeseer, and Pubmed are smaller graphs, with the OGB datasets having more nodes and edges. We adopt the single fixed train/validation/test split with percentages 85/5/10% for Cora, Citeseer, and Pubmed. For OGB datasets, we use the fixed splits provided by the OGB benchmark [34].

### C.2 Experimental Settings

**Training Settings**. We use the binary cross entropy loss to train each model. The loss is optimized using the Adam optimizer [44]. During training we randomly sample one negative sample per positive sample. Each model is trained for a maximum of 9999 epochs, with the process set to terminate when there are no improvements observed in the validation performance over $n$ checkpoints. The choice of $n$ is influenced by both the specific dataset and the complexity of the model. For smaller datasets, such as Cora, Citeseer, and Pubmed, we set $n = 50$ uniformly across models (except for

NBFNet and SEAL where $n = 20$ due to their computational inefficiency). When training on larger OGB datasets, we use a stratified approach: $n = 100$ for the simpler methods (i.e., the embedding and GNN-based models) and $n = 20$ for the more advanced methods. This is due to the increased complexity and runtime of more advanced methods. An exception is for ogbl-citation2, the largest dataset. To accommodate for its size, we limit the maximum number of epochs to the recommended value from each model's source code. Furthermore, we set $n = 20$ for all models.

In order to accommodate the computational requirements for our extensive experiments, we harness a variety of high-capacity GPU resources. This includes: Tesla V100 32Gb, NVIDIA RTX A6000 48Gb, NVIDIA RTX A5000 24Gb, and Quadro RTX 8000 48Gb.

Table 8: Hyperparameter Search Ranges

| Dataset | Learning Rate | Dropout | Weight Decay | # Model Layers | # Prediction Layers | Embedding Dim |
|---|---|---|---|---|---|---|
| Cora | (0.01, 0.001) | (0.1, 0.3, 0.5) | (1e-4, 1e-7, 0) | (1, 2, 3) | (1, 2, 3) | (128, 256) |
| Citeseer | (0.01, 0.001) | (0.1, 0.3, 0.5) | (1e-4, 1e-7, 0) | (1, 2, 3) | (1, 2, 3) | (128, 256) |
| Pubmed | (0.01, 0.001) | (0.1, 0.3, 0.5) | (1e-4, 1e-7, 0) | (1, 2, 3) | (1, 2, 3) | (128, 256) |
| ogbl-collab | (0.01, 0.001) | (0, 0.3, 0.5) | 0 | 3 | 3 | 256 |
| ogbl-ddi | (0.01, 0.001) | (0, 0.3, 0.5) | 0 | 3 | 3 | 256 |
| ogbl-ppa | (0.01, 0.001) | (0, 0.3, 0.5) | 0 | 3 | 3 | 256 |
| ogbl-citation2 | (0.01, 0.001) | (0, 0.3, 0.5) | 0 | 3 | 3 | 128 |

**Hyperparameter Settings**. We present the hyparameter searching range in Table 8. For the smaller graphs, Cora, Citeseer, and Pubmed, we have a larger search space. However, it's not feasible to tune over such large space for OGB datasets. By following the most commonly used settings among published hyperparameters, we fix the weight decay, number of model and prediction layers, and the embedding dimension. Furthermore, due to GPU memory constraints, the embedding size is reduced to be 128 for the largest dataset ogbl-citation2.

We note that several exceptions exist to these ranges when they result in significant performance degradations. In such instances, adjustments are guided by the optimal hyperparameters published in the respective source codes. This includes:

- PEG [33]: Adhering to the optimal hyperparameters presented in the source code,[4] when training on ogbl-ddi we set the number of model layers to 2 and the maximum number of epochs to 400.

- NCN/NCNC [15]: When training on ogbl-ddi, we adhere to the suggested optimal hyperparameters used in the source code.[5] Specifically, we set the number of model layers to be 1, and we don't apply the pretraining for NCNC to facilitate a fair comparison.

- NBFNet [19]: Due to the expensive nature of NBFNet, we further fix the weight decay to 0 when training on Cora, Citeseer, and Pubmed. Furthermore, we follow the suggested hyperparameters [6] and set the embedding dimension to be 32 and the number of model layers to be 6.

- SEAL [12]: Due to the computational inefficiency of SEAL, when training on Cora, Citeseer and Pubmed we further fix the weight decay to 0. Furthermore, we adhere to the published hyperparameters [7] and fix the number of model layers to be 3 and the embedding dimension to be 256.

- BUDDY [13]: When training on ogbl-ppa, we incorporate the RA and normalized degree as input features while excluding the raw node features. This is based on the optimal hyperparameters published by the authors.[8]

# D    Reported vs. Our Results on ogbl-ddi

In Section 3 (see observation 2), we remarked that there is divergence between the reported results and our results on ogbl-ddi for some methods. A comprehensive comparison of this discrepancy is

---

[4]https://github.com/Graph-COM/PEG/

[5]https://github.com/GraphPKU/NeuralCommonNeighbor/

[6]https://github.com/DeepGraphLearning/NBFNet/

[7]https://github.com/facebookresearch/SEAL_OGB/

[8]https://github.com/melifluos/subgraph-sketching/

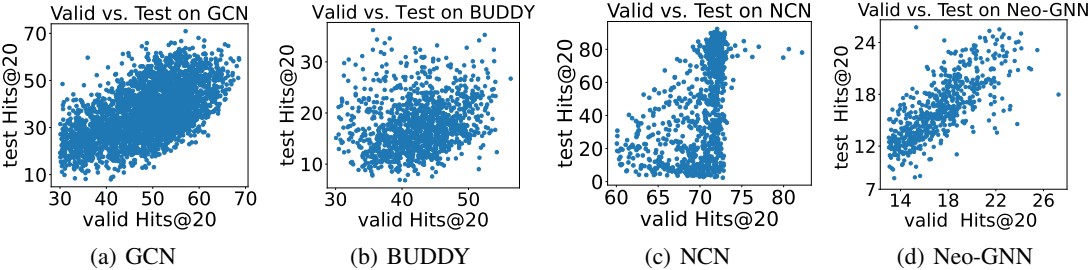

Figure 6: Validation vs. test performance for GCN, BUDDY, NCN and Neo-GNN on ogbl-ddi under the existing evaluation setting.

shown in Table 9. The reported results for Node2Vec, MF, GCN, and SAGE are taken from [34]. The results for the other methods are from their original paper: SEAL [26], BUDDY [13], Neo-GNN [14], NCN [15], NCNC [15], and PEG [33].

Table 9: Comparison results between ours and reported results on ogbl-ddi (Hits@20).

|  | Node2Vec | MF | GCN | SAGE | SEAL | BUDDY | Neo-GNN | NCN | NCNC | PEG |
|---|---|---|---|---|---|---|---|---|---|---|
| Reported | 23.26 ± 2.09 | 13.68 ± 4.75 | 37.07 ± 5.07 | **53.90 ± 4.74** | **30.56 ± 3.86** | **78.51 ± 1.36** | **63.57 ± 3.52** | **82.32 ± 6.10** | **84.11 ± 3.67** | **43.80 ± 0.32** |
| Ours | **34.69 ± 2.90** | **23.50 ± 5.35** | **49.90 ± 7.23** | 49.84 ± 15.56 | 25.25 ± 3.90 | 29.60 ± 4.75 | 20.95 ± 6.03 | 76.52 ± 10.47 | 70.23 ± 12.11 | 30.28 ± 4.92 |

# E    Additional Investigation on ogbl-ddi

In this section, we present the additional investigation on the ogbl-ddi dataset. In Section E.1 we examine under the existing evaluation setting, there exists a poor relationship between the validation and test performance on ogbl-ddi for many methods. We then demonstrate in Section E.2 that under HeaRT this problem is lessened.

## E.1    Existing Evaluation Setting

Upon inspection, we found that there is a poor relationship between the validation and test performance on ogbl-ddi. Since we choose the best hyperparameters based on the validation set, this makes it difficult to properly tune any model on ogbl-ddi. To demonstrate this point, we record the validation and test performance at multiple checkpoints during the training process. The experiments are conducted over 10 seeds. To ensure that our results are not caused by our hyperparameter settings, we use the reported hyperparameters for each model. Lastly, we plot the results for GCN, BUDDY, NCN, and Neo-GNN in Figure 6. It is clear from the results that there exists a poor relationship between the validation and test performance. For example, for NCN, a validation performance of 70 can imply a test performance of 3 to 80. Further investigation is needed to uncover the cause of this misalignment.

## E.2    New Evaluation Setting

Under our new setting, we find that the validation and test performance have a much better relationship. In Section E.1 we observed that there exists a poor relationship between the validation and test performance on ogbl-ddi under the existing evaluation setting. This meddles with our ability to choose the best hyperparameters for each model, as good validation performance is not indicative of good test performance. However, this does not seem to be the case under the new evaluation setting. In Figure 7 we plot the relationship between the validation and test performance by checkpoint for various models. Compared to the same plots under the existing setting (Figure 6), the new results display a much better relationship.

While it's unclear what is the cause of the poor relationship between the test and validation performance under the existing setting, we conjecture that tailoring the negatives to each positive sample allows for a more natural comparison between a positive sample and its negatives. This may help

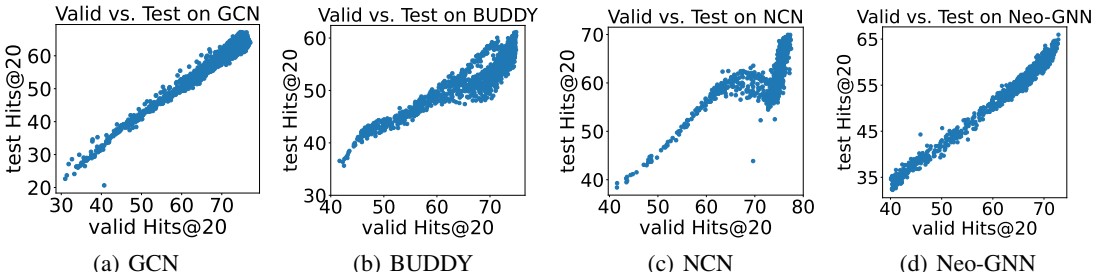

| (a) GCN | (b) BUDDY | (c) NCN | (d) Neo-GNN |

Figure 7: Validation vs. test performance when utilizing HeaRT for GCN, BUDDY, NCN and Neo-GNN on ogbl-ddi. We find they have a much stronger relationship than under the existing setting (see Figure 6).

Table 10: Additional results on Cora(%) under the existing evaluation setting. Highlighted are the results ranked **first**, **second**, and **third**.

| | Models | Hits@1 | Hits@3 | Hits@10 | Hits@100 |
|---|---|---|---|---|---|
| | CN | 13.47 | 13.47 | 42.69 | 42.69 |
| | AA | 22.2 | 39.47 | 42.69 | 42.69 |
| Heuristic | RA | 20.11 | 39.47 | 42.69 | 42.69 |
| | Shortest Path | 0 | 0 | 42.69 | 71.35 |
| | Katz | 19.17 | 28.46 | 51.61 | 74.57 |
| | Node2Vec | **22.3 ± 11.76** | **41.63 ± 10.5** | 62.34 ± 2.35 | 84.88 ± 0.96 |
| Embedding | MF | 7.76 ± 5.61 | 13.26 ± 4.52 | 29.16 ± 6.68 | 66.39 ± 5.03 |
| | MLP | 18.79 ± 11.40 | 35.35 ± 10.71 | 53.59 ± 3.57 | 85.52 ± 1.44 |
| | GCN | 16.13 ± 11.18 | 32.54 ± 10.83 | 66.11 ± 4.03 | 91.29 ± 1.25 |
| GNN | GAT | 18.02 ± 8.96 | **42.28 ± 6.37** | 63.82 ± 2.72 | 90.70 ± 1.03 |
| | SAGE | **29.01 ± 6.42** | **44.51 ± 6.57** | 63.66 ± 4.98 | 91.00 ± 1.52 |
| | GAE | 17.57 ± 4.37 | 24.82 ± 4.91 | **70.29 ± 2.75** | **92.75 ± 0.95** |
| | SEAL | 12.35 ± 8.57 | 38.63 ± 4.96 | 55.5 ± 3.28 | 84.76 ± 1.6 |
| | BUDDY | 12.62 ± 6.69 | 29.64 ± 5.71 | 59.47 ± 5.49 | 91.42 ± 1.26 |
| | Neo-GNN | 4.53 ± 1.96 | 33.36 ± 9.9 | 64.1 ± 4.31 | 87.76 ± 1.37 |
| GNN+heuristic | NCN | 19.34 ± 9.02 | 38.39 ± 7.01 | **74.38 ± 3.15** | **95.56 ± 0.79** |
| | NCNC | 9.79 ± 4.56 | 34.31 ± 8.87 | **75.07 ± 1.95** | **95.62 ± 0.84** |
| | NBFNet | **29.94 ± 5.78** | 38.29 ± 3.03 | 62.79 ± 2.53 | 88.63 ± 0.46 |
| | PEG | 5.88 ± 1.65 | 30.53 ± 6.42 | 62.49 ± 4.05 | 91.42 ± 0.8 |

produce more stable evaluation metrics, thereby strengthening the alignment between the validation and test performance.

# F  Additional Results Under the Existing Setting

We present additional results of Cora, Citeseer, Pubmed and OGB datasets in Tables 10-13 under the existing setting. We also omit the MRR for ogbl-collab, ogbl-ddi, and ogbl-ppa. This is because the large number of negative samples make it very inefficient to calculate.

# G  Additional Details on HeaRT

As described in Section 4.2, given a positive sample $(a, b)$, we seeks to generate $K$ negative samples to evaluate against. The negative samples are drawn from the set of possible corruptions of $(a, b)$, i.e, $S(a, b)$ (see Eq. (3)). Multiple heuristics are used to determine which $K$ negative samples to use. Furthermore, the negative samples are split evenly between both nodes. That is, we generate $K/2$ negative samples that contain either node $a$ and $b$, respectively. This process is illustrated in Figure 2.

The rest of this section is structured as follows. In Section G.1 we describe how we use multiple heuristics for estimating the difficulty of negative samples. Then in Section G.2 we describe how we combine the ranks given by different heuristic methods.

Table 11: Additional results on Citeseer(%) under the existing evaluation setting. Highlighted are the results ranked **first**, **second**, and **third**.

| | Models | Hits@1 | Hits@3 | Hits@10 | Hits@100 |
|---|---|---|---|---|---|
| Heuristic | CN | 13.85 | 35.16 | 35.16 | 35.16 |
| | AA | 21.98 | 35.16 | 35.16 | 35.16 |
| | RA | 18.46 | 35.16 | 35.16 | 35.16 |
| | Shortest Path | 0 | 53.41 | 56.92 | 62.64 |
| | Katz | 24.18 | 54.95 | 57.36 | 62.64 |
| Embedding | Node2Vec | 30.24 ± 16.37 | 54.15 ± 6.96 | 68.79 ± 3.05 | 89.89 ± 1.48 |
| | MF | 19.25 ± 6.71 | 29.03 ± 4.82 | 38.99 ± 3.26 | 59.47 ± 2.69 |
| | MLP | 30.22 ± 10.78 | 56.42 ± 7.90 | 69.74 ± 2.19 | 91.25 ± 1.90 |
| GNN | GCN | 37.47 ± 11.30 | 62.77 ± 6.61 | 74.15 ± 1.70 | 91.74 ± 1.24 |
| | GAT | 34.00 ± 11.14 | 62.72 ± 4.60 | 74.99 ± 1.78 | 91.69 ± 2.11 |
| | SAGE | 27.08 ± 10.27 | 65.52 ± 4.29 | 78.06 ± 2.26 | **96.50 ± 0.53** |
| | GAE | **54.06 ± 5.8** | 65.3 ± 2.54 | **81.72 ± 2.62** | 95.17 ± 0.5 |
| GNN+heuristic | SEAL | 31.25 ± 8.11 | 46.04 ± 5.69 | 60.02 ± 2.34 | 85.6 ± 2.71 |
| | BUDDY | **49.01 ± 15.07** | **67.01 ± 6.22** | **80.04 ± 2.27** | 95.4 ± 0.63 |
| | Neo-GNN | 41.01 ± 12.47 | 59.87 ± 6.33 | 69.25 ± 1.9 | 89.1 ± 0.97 |
| | NCN | 35.52 ± 13.96 | 66.83 ± 4.06 | 79.12 ± 1.73 | **96.17 ± 1.06** |
| | NCNC | **53.21 ± 7.79** | **69.65 ± 3.19** | **82.64 ± 1.4** | **97.54 ± 0.59** |
| | NBFNet | 17.25 ± 5.47 | 51.87 ± 2.09 | 68.97 ± 0.77 | 86.68 ± 0.42 |
| | PEG | 39.19 ± 8.31 | **70.15 ± 4.3** | 77.06 ± 3.53 | 94.82 ± 0.81 |

Table 12: Additional results on Pubmed(%) under the existing evaluation setting. Highlighted are the results ranked **first**, **second**, and **third**.

| | Models | Hits@1 | Hits@3 | Hits@10 | Hits@100 |
|---|---|---|---|---|---|
| Heuristic | CN | 7.06 | 12.95 | 27.93 | 27.93 |
| | AA | 12.95 | 16 | 27.93 | 27.93 |
| | RA | 11.67 | 15.21 | 27.93 | 27.93 |
| | Shortest Path | 0 | 0 | 27.93 | 60.36 |
| | Katz | 12.88 | 25.38 | 42.17 | 61.8 |
| Embedding | Node2Vec | **29.76 ± 4.05** | 34.08 ± 2.43 | 44.29 ± 2.62 | 63.07 ± 0.34 |
| | MF | 12.58 ± 6.08 | 22.51 ± 5.6 | 32.05 ± 2.44 | 53.75 ± 2.06 |
| | MLP | 7.83 ± 6.40 | 17.23 ± 2.79 | 34.01 ± 4.94 | 84.19 ± 1.33 |
| GNN | GCN | 5.72 ± 4.28 | 19.82 ± 7.59 | 56.06 ± 4.83 | 87.41 ± 0.65 |
| | GAT | 6.45 ± 10.37 | 23.02 ± 10.49 | 46.77 ± 4.03 | 80.95 ± 0.72 |
| | SAGE | 11.26 ± 6.86 | 27.23 ± 7.48 | 48.18 ± 4.60 | **90.02 ± 0.70** |
| | GAE | 1.99 ± 0.12 | 31.75 ± 1.13 | 45.48 ± 1.07 | 84.3 ± 0.31 |
| GNN+heuristic | SEAL | **30.93 ± 8.35** | **40.58 ± 6.79** | 48.45 ± 2.67 | 76.06 ± 4.12 |
| | BUDDY | 15.31 ± 6.13 | 29.79 ± 6.76 | 46.62 ± 4.58 | 83.21 ± 0.59 |
| | Neo-GNN | 19.95 ± 5.86 | 34.85 ± 4.43 | **56.25 ± 3.42** | 86.12 ± 1.18 |
| | NCN | 26.38 ± 6.54 | **36.82 ± 6.56** | **62.15 ± 2.69** | **90.43 ± 0.64** |
| | NCNC | 9.14 ± 5.76 | 33.01 ± 6.28 | **61.89 ± 3.54** | **91.93 ± 0.6** |
| | NBFNet | **40.47 ± 2.91** | **44.7 ± 2.58** | 54.51 ± 0.84 | 79.18 ± 0.71 |
| | PEG | 8.52 ± 3.73 | 24.46 ± 6.94 | 45.11 ± 4.02 | 76.45 ± 3.83 |

Table 13: Additional results on OGB datasets(%) under the existing evaluation setting. Highlighted are the results ranked **first**, **second**, and **third**.

| | ogbl-collab | | ogbl-ddi | | ogbl-ppa | | ogbl-citation2 | | |
|---|---|---|---|---|---|---|---|---|---|
| | Hits@20 | Hits@100 | Hits@50 | Hits@100 | Hits@20 | Hits@50 | Hits@20 | Hits@50 | Hits@100 |
| CN | 49.98 | 65.6 | 26.51 | 34.52 | 13.26 | 19.67 | 77.99 | 77.99 | 77.99 |
| AA | 55.79 | 65.6 | 27.07 | 36.35 | 14.96 | 21.83 | 77.99 | 77.99 | 77.99 |
| RA | 55.01 | 65.6 | 19.14 | 31.17 | 25.64 | **38.81** | 77.99 | 77.99 | 77.99 |
| Shortest Path | 46.49 | 66.82 | 0 | 0 | 0 | 0 | >24h | >24h | >24h |
| Katz | **58.11** | **71.04** | 26.51 | 34.52 | 13.26 | 19.67 | 78 | 78 | 78 |
| Node2Vec | 40.68 ± 1.75 | 55.58 ± 0.77 | 59.19 ± 3.61 | 73.49 ± 3.18 | 11.22 ± 1.91 | 19.22 ± 1.69 | 82.8 ± 0.13 | 92.33 ± 0.1 | 96.44 ± 0.03 |
| MF | 39.99 ± 1.25 | 43.22 ± 1.94 | 45.51 ± 11.13 | 61.72 ± 6.56 | 9.33 ± 2.83 | 21.08 ± 3.92 | 70.8 ± 12.0 | 74.48 ± 10.42 | 75.5 ± 10.13 |
| MLP | 27.66 ± 1.61 | 42.13 ± 1.09 | N/A | N/A | 0.16 ± 0.0 | 0.26 ± 0.03 | 74.16 ± 0.1 | 86.59 ± 0.08 | 93.14 ± 0.06 |
| GCN | 44.92 ± 3.72 | 62.67 ± 2.14 | 74.54 ± 4.74 | 85.03 ± 3.41 | 11.17 ± 2.93 | 21.04 ± 3.11 | **98.01 ± 0.04** | **99.03 ± 0.02** | **99.48 ± 0.02** |
| GAT | 43.59 ± 4.17 | 62.24 ± 2.29 | 55.46 ± 10.16 | 69.74 ± 10.01 | OOM | OOM | OOM | OOM | OOM |
| SAGE | 50.77 ± 2.33 | 65.36 ± 1.05 | **93.48 ± 1.36** | **97.37 ± 0.55** | 19.37 ± 2.65 | 31.3 ± 2.36 | 97.48 ± 0.03 | 98.75 ± 0.03 | 99.3 ± 0.02 |
| GAE | OOM | OOM | 12.39 ± 8.74 | 14.03 ± 9.22 | OOM | OOM | OOM | OOM | OOM |
| SEAL | 54.19 ± 1.57 | 69.94 ± 0.72 | 43.34 ± 3.23 | 52.2 ± 1.78 | 21.81 ± 4.3 | 36.88 ± 4.06 | 94.61 ± 0.11 | 95.0 ± 0.12 | 95.37 ± 0.14 |
| BUDDY | **57.78 ± 0.59** | 67.87 ± 0.87 | 53.36 ± 2.57 | 71.04 ± 2.56 | **26.33 ± 2.63** | 38.18 ± 1.32 | **97.79 ± 0.07** | **98.86 ± 0.04** | 99.38 ± 0.03 |
| Neo-GNN | **57.05 ± 1.56** | **71.76 ± 0.55** | 33.88 ± 10.1 | 46.55 ± 13.29 | 26.16 ± 1.24 | 37.95 ± 1.45 | 97.05 ± 0.07 | 98.75 ± 0.03 | **99.41 ± 0.02** |
| NCN | 50.27 ± 2.72 | 67.58 ± 0.09 | **95.51 ± 0.87** | **97.54 ± 0.7** | **40.29 ± 2.22** | **53.35 ± 1.77** | **97.97 ± 0.03** | **99.02 ± 0.02** | **99.5 ± 0.01** |
| NCNC | 54.91 ± 2.84 | **70.91 ± 0.25** | **92.34 ± 2.42** | **96.35 ± 0.52** | **40.1 ± 1.06** | **52.09 ± 1.99** | 97.22 ± 0.78 | 98.2 ± 0.71 | 98.77 ± 0.6 |
| NBFNet | OOM | OOM | >24h | >24h | OOM | OOM | OOM | OOM | OOM |
| PEG | 33.57 ± 7.40 | 55.14 ± 2.10 | 47.93 ± 3.18 | 59.95 ± 2.52 | OOM | OOM | OOM | OOM | OOM |

## G.1 Determining Hard Negative Samples

We are first tasked with *how to choose the negative samples*. As discussed and shown in Section 4.2, we want to select the negative samples from $S(a, b)$ such that they are non-trivial to classify. Hence, as inspired by the candidate generation process in real-world recommender systems [39, 40], we aim to select a set of 'hard' negative samples that are more relevant to the source node. The candidate generation process is typically based on some primitive and simple link prediction heuristics. These heuristics can be also treated as link prediction methods (see Tables 1 and 2).

We use multiple heuristics that capture a variety of different information. Most link prediction heuristics can be categorized into two main categories: local heuristics and global heuristics [1]. Local heuristics attempt to capture the local neighborhood information that exists near the node pair while global heuristics attempt to use the whole graph structure. To capture the *local* information we use resource allocation (RA) [29], a CN-based approach. Existing results show that RA can achieve strong performance on most datasets (see Tables 1 and 2). To measure the *global* information we use the personalized pagerank score (PPR) [41]. Random walk based methods are commonly used for candidate generation [39, 40]. Lastly, we further include the cosine feature similarity for the Cora, Citeseer, and Pubmed datasets. This is due to the strong performance of a MLP on those datasets. By combining these heuristics, we are able to generate a diverse set of negative samples for each positive sample.

For each heuristic we then rank all the possible negative samples. We first denote the score of a heuristic $i$ for a pair of nodes $a$ and $b$ as $h_i(a, b)$. Let's say we want to rank all negative samples that contain a node $a$, i.e., $(a, *)$. The rank across all nodes is given by:

$$R_i = \underset{v \in \bar{V}}{\text{ArgSort}}\, h_i(a, v), \tag{5}$$

where $R_i$ denotes the ranking for heuristic $i$ and and $\bar{V}$ is a subset of the set of nodes in the graph $V$. We note that all training samples, self-loops, and the sample itself are not able to be chosen as negative samples. Furthermore, when choosing negative samples for the test samples, we disallow validation samples to be chosen as well. As such, we only consider a subset of nodes $\bar{V} \in V$. This is analogous to the filtered setting used in KGC [38].

We now have all possible negative samples ranked according to multiple heuristics. However, it is unclear how to choose the negative samples from multiple ranked lists. In the next subsection we detail how we combine the ranks according to each heuristic. This will give us a final ranking, of which we can choose the top $K/2$ as the negative samples for that node.

## G.2 Combining Heuristic Ranks

In this subsection we tackle the problem of combing the negative sample ranks given by multiple heuristics. More concretely, say we use $m$ heuristics and rank all the samples according to each. We want to arrive at a combined ranking $R_{\text{total}}$ that is composed of each rank,

$$R_{\text{total}} = \phi(R_1, R_2, \cdots, R_m). \tag{6}$$

We model $\phi$ via Borda's method [45]. Let $R_i(a, v)$ be the rank of the node pair $(a, v)$ for heuristic $i$. The combined rank $R_{\text{total}}(a, v)$ across $m$ ranked lists is given by:

$$R_{\text{total}}(a, v) = g\left(R_1(a, v), R_2(a, v), \cdots, R_m(a, v)\right), \tag{7}$$

where $g$ is an aggregation function. We set $g = \min(\cdot)$. This is done as it allows us to capture a more distinct set of samples by selecting the "best" for each heuristic. This is especially true when there is strong disagreement between the different heuristics. A final ranking is then done on $R_{\text{total}}$ to select the top nodes,

$$R_f = \underset{v \in \bar{V}}{\text{ArgSort}}\, R_{\text{total}}, \tag{8}$$

where $R_f$ is the final ranking. The highest $K/2$ nodes are then selected from $R_f$. Lastly, we note that for some nodes there doesn't exist sufficient scores to rank $K/2$ total nodes. In this case the remaining nodes are chosen randomly. The full generation process for a node $a$ is detailed in Algorithm 1.

**Algorithm 1** Generating Negative Samples of Form $(a, *)$

**Require:**
    $a$ = Node to generate samples for
    $\bar{V}$ = Possible nodes to use for negative samples
    $\mathcal{H} = \{h_1, h_2, \cdots, h_m\}$                         ▷ Set of $m$ heuristics

1: **for** $i \in |\mathcal{H}|$ **do**
2:      $R_i = \underset{v \in \bar{V}}{\text{ArgSort}}\, h_i(a, v)$                   ▷ Sort by each heuristic individually
3: **end for**
4: **for** $v \in \bar{V}$ **do**
5:      $R_{\text{total}}(a, v) = \min\left(R_1(a, v), R_2(a, v), \cdots, R_m(a, v)\right)$      ▷ Combine the rankings
6: **end for**

7: $R_f = \underset{v \in \bar{V}}{\text{ArgSort}}\, R_{\text{total}}(a, v)$                   ▷ Sort by combined ranking
8: **return** $R_f[: K/2]$                        ▷ Return the top K/2 ranked nodes

Table 14: Results on ogbl-collab under HeaRT with new hard negative samples. Highlighted are the results ranked **first**, **second**, and **third**.

|  | MRR | Hits20 | Hits50 | Hits100 |
|---|---|---|---|---|
| CN | 4.20 | 16.46 | 30.52 | 42.80 |
| AA | 5.07 | 19.59 | 33.74 | 45.20 |
| RA | **6.29** | **24.29** | **36.68** | 46.42 |
| Shortest Path | 2.66 | 15.98 | 33.77 | 45.85 |
| Katz | **6.31** | **24.34** | **39.18** | 48.80 |
| Node2Vec | 4.68 ± 0.08 | 16.84 ± 0.17 | 28.56 ± 0.17 | 41.84 ± 0.25 |
| MF | 4.89 ± 0.25 | 18.86 ± 0.40 | 30.83 ± 0.22 | 43.23 ± 0.34 |
| MLP | 5.37 ± 0.14 | 16.15 ± 0.27 | 28.88 ± 0.32 | 46.83 ± 0.33 |
| GCN | 6.09 ± 0.38 | 22.48 ± 0.81 | 35.29 ± 0.49 | **50.83 ± 0.21** |
| GAT | 4.18 ± 0.33 | 18.30 ± 1.42 | 32.92 ± 1.41 | 46.71 ± 0.84 |
| SAGE | 5.53 ± 0.5 | 21.26 ± 1.32 | 33.48 ± 1.40 | 48.33 ± 0.49 |
| GAE | OOM | OOM | OOM | OOM |
| SEAL | **6.43 ± 0.32** | 21.57 ± 0.38 | 33.57 ± 0.84 | 43.06 ± 1.09 |
| Neo-GNN | 5.23 ± 0.9 | 21.03 ± 3.39 | 36.11 ± 2.36 | **49.25 ± 0.81** |
| NBFNet | OOM | OOM | OOM | OOM |
| BUDDY | 5.67 ± 0.36 | **23.35 ± 0.73** | **39.04 ± 0.11** | **50.49 ± 0.09** |
| PEG | 4.83 ± 0.21 | 18.29 ± 1.06 | 30.12 ± 0.63 | 45.40 ± 0.66 |
| NCN | 5.09 ± 0.38 | 20.84 ± 1.31 | 34.53 ± 0.98 | 45.69 ± 0.42 |
| NCNC | 4.73 ± 0.86 | 20.49 ± 3.97 | 34.96 ± 3.80 | 46.93 ± 2.04 |

# H   Additional Investigation on ogbl-collab

In Section 4.3 we observed that under HeaRT, both the Shortest Path and Katz perform considerably well on ogbl-collab under the HeaRT setting. Specifically, the MRR gap between the second-ranked method (Shortest Path) and the third (BUDDY) is 14.29. Since we do not observe this under the existing setting, we further investigate the reason.

We analyzed the performance of both methods on the ogbl-collab dataset. Interestingly, we find that it is due to the dataset being a dynamic graph. In this dataset, nodes represent authors and edges represent a collaboration between two authors. Each edge further includes an attribute that specifies the year of collaboration. Specifically, each edge takes the form of (Author 1, Year, Author 2). The task is to predict collaborations in 2019 (test) based on those until 2017 (training) and 2018 (validation).

We found that of the positive samples in the test set, around $46\%$ also appear as positive samples in the training set. In particular, an edge (Author 1, 2017, Author 2) in the training data may also "appear" in the test data in the form of (Author 1, 2019, Author 2). This is because two authors who collaborated in the past often tend to collaborate again in the future. As such, when evaluating the test sample (Author 1, 2019, Author 2), there is a path of length 1 between the two authors. Furthermore, this co-occurence phenomenon is common among positive samples but not observed among negatives. This is because we exclude the positive training samples when generating the

Table 15: Additional results on Cora, Citeseer, and Pubmed(%) under HeaRT. Highlighted are the results ranked first, second, and third.

| Models | Cora | | | Citeseer | | | Pubmed | | |
|---|---|---|---|---|---|---|---|---|---|
| | Hits1 | Hits3 | Hits100 | Hits1 | Hits3 | Hits100 | Hits1 | Hits3 | Hits100 |
| CN | 3.98 | 10.25 | 38.71 | 2.2 | 9.45 | 33.63 | 0.47 | 1.49 | 19.29 |
| AA | 5.31 | 12.71 | 38.9 | 3.96 | 12.09 | 33.85 | 0.74 | 1.87 | 20.37 |
| RA | 5.31 | 12.52 | 38.52 | 4.18 | 11.21 | 34.07 | 0.72 | 1.78 | 20.04 |
| Shortest Path | 0.57 | 2.85 | 55.6 | 0.22 | 3.52 | 53.85 | 0 | 0.02 | 21.57 |
| Katz | 4.64 | 11.95 | 59.96 | 3.74 | 11.87 | 55.82 | 0.74 | 2.12 | 32.78 |
| Node2Vec | 5.69 ± 0.81 | 15.1 ± 0.99 | 77.21 ± 2.34 | 9.63 ± 0.82 | 23.5 ± 1.37 | 84.46 ± 1.86 | 0.75 ± 0.14 | 2.4 ± 0.32 | 52.27 ± 0.65 |
| MF | 1.46 ± 0.8 | 5.46 ± 1.67 | 59.68 ± 3.41 | 2.68 ± 0.92 | 7.25 ± 0.98 | 53.25 ± 2.91 | 1.13 ± 0.24 | 3.25 ± 0.44 | 50.56 ± 1.11 |
| MLP | 5.48 ± 0.99 | 14.15 ± 1.56 | 77.0 ± 1.02 | 10.44 ± 0.82 | 26.46 ± 1.24 | 86.83 ± 1.36 | 1.28 ± 0.22 | 4.33 ± 0.28 | 76.34 ± 0.79 |
| GCN | **7.59 ± 0.61** | **17.46 ± 0.82** | **85.47 ± 0.52** | 9.27 ± 0.99 | 23.19 ± 0.98 | 89.1 ± 2.13 | 2.09 ± 0.31 | 5.58 ± 0.27 | 73.59 ± 0.53 |
| GAT | 5.03 ± 0.81 | 13.66 ± 0.67 | 80.87 ± 1.32 | 8.02 ± 1.21 | 20.09 ± 0.82 | 86.83 ± 1.09 | 1.14 ± 0.16 | 3.06 ± 0.36 | 67.06 ± 0.69 |
| SAGE | 5.48 ± 0.97 | **15.43 ± 1.07** | 81.61 ± 0.96 | 8.37 ± 1.62 | 23.74 ± 1.62 | **92.33 ± 0.68** | **3.03 ± 0.46** | **8.19 ± 1.0** | **79.47 ± 0.53** |
| GAE | **9.72 ± 0.73** | **19.24 ± 0.76** | 79.66 ± 0.95 | **13.81 ± 0.82** | **27.71 ± 1.34** | 85.49 ± 1.37 | 1.48 ± 0.23 | 4.05 ± 0.39 | 59.79 ± 0.67 |
| SEAL | 3.89 ± 2.04 | 10.82 ± 4.04 | 61.9 ± 13.97 | 5.08 ± 1.31 | 13.68 ± 1.32 | 68.94 ± 2.3 | 1.47 ± 0.32 | 4.71 ± 0.68 | 65.81 ± 2.43 |
| BUDDY | 5.88 ± 0.60 | 13.76 ± 1.03 | 82.46 ± 1.79 | 10.09 ± 0.50 | 26.11 ± 1.26 | **92.66 ± 0.92** | **2.24 ± 0.17** | 5.93 ± 0.21 | 72.01 ± 0.46 |
| Neo-GNN | 5.71 ± 0.41 | 13.89 ± 0.82 | 80.28 ± 1.08 | 6.81 ± 0.73 | 17.8 ± 1.19 | 85.51 ± 1.01 | 1.90 ± 0.24 | **6.07 ± 0.47** | **76.57 ± 0.58** |
| NCN | 4.85 ± 0.81 | 14.46 ± 0.98 | **84.14 ± 1.24** | **16.77 ± 2.05** | **30.51 ± 0.97** | 90.42 ± 0.98 | 1.13 ± 0.18 | 3.95 ± 0.24 | 71.46 ± 0.97 |
| NCNC | 4.78 ± 0.71 | 14.72 ± 1.24 | **85.62 ± 0.83** | **11.14 ± 0.82** | **27.21 ± 0.96** | **92.73 ± 1.16** | **2.73 ± 0.49** | **7.05 ± 0.72** | **79.22 ± 0.96** |
| NBFNet | 5.31 ± 1.16 | 14.95 ± 0.72 | 76.24 ± 0.68 | 5.95 ± 1.06 | 14.53 ± 1.19 | 72.66 ± 0.95 | >24h | >24h | >24h |
| PEG | **6.98 ± 0.57** | 14.93 ± 0.61 | 82.52 ± 1.28 | 9.93 ± 0.6 | 21.91 ± 0.59 | 90.15 ± 1.43 | 0.88 ± 0.18 | 2.61 ± 0.39 | 64.95 ± 1.81 |

negative samples for evaluation. **As a result of this exclusion, the presence of a direct link, or shortest path, between two authors tends to suggest a positive sample, while its absence often corresponds to a negative sample.** This explains why methods like Shortest Path and Katz can achieve good performance on ogbl-collab.

To mitigate this issue, we introduce a **new set of hard negatives**, permitting the negative samples to also exhibit a shortest path of 1. We do so by not excluding the positive training samples when generating the negative samples for validation and test. We note that we also do not exclude positive validation edges when generating the negatives for test. In simpler terms, when creating negative samples for testing, both positive samples from training and validation are considered. This means that negative samples during testing could present in the training and validation positive samples. This approach is reasonable and well-aligned with the real-world scenario in the context of collaboration graphs. Specifically, authors who collaborated in the past might not do so in the future. For instance, just because the positive sample (Author 2, 2017, Author 3) exists, it does not imply that (Author 2, 2019, Author 3) is also true. However, our previous setting assumed so.

Results using the new hard negatives are presented in Table 14. We observe that the Shortest Path no longer demonstrates the best performance. In contrast, both Katz and RA perform notably well, underscoring the significance of the common neighbors information (i.e., paths of length 2). Since Katz considers all paths between two nodes, it might benefit from the effective performance of methods based on common neighbors. Additionally, the overall results are inferior to those using the old hard negatives, as seen in Table 5. For instance, the MRR values in Table 5 exceed 10, the highest MRR in Table 14 is just approximately 6, suggesting the increased difficulty introduced by the new hard negative strategy.

# I  Additional Results Under HeaRT

We present additional results of Cora, Citeseer, Pubmed, and OGB datasets under HeaRT in Table 15 and Table 16. These results include other hit metrics not found in the main tables.

# J  Dataset Licenses

The license for each dataset can be found in Table 17.

# K  Limitation

One potential limitation of HeaRT lies in the generation of customized negative samples for each positive sample. This design may result in an increased number of negative samples compared to the existing setting. Although this provides a more realistic evaluation, it could have an impact on the

Table 16: Additional results on OGB datasets(%) under HeaRT. Highlighted are the results ranked **first**, **second**, and **third**.

| Models | ogbl-collab | | ogbl-ddi | | ogbl-ppa | | ogbl-citation2 | |
| --- | --- | --- | --- | --- | --- | --- | --- | --- |
| | Hits50 | Hits100 | Hits50 | Hits100 | Hits50 | Hits100 | Hits50 | Hits100 |
| CN | 38.39 | 47.4 | 70.12 | 86.53 | 80.53 | 86.51 | 57.56 | 68.04 |
| AA | 42.61 | 50.25 | 71.08 | 87.36 | 81.93 | 87.55 | 58.87 | 69.39 |
| RA | 46.9 | 51.78 | 76.39 | 90.96 | 81.65 | 86.84 | 58.88 | 68.83 |
| Shortest Path | 46.97 | 48.11 | 0 | 0 | 1.34 | 1.4 | >24h | >24h |
| Katz | 51.07 | 54.28 | 70.12 | 86.53 | 80.53 | 86.51 | 54.97 | 67.56 |
| Node2Vec | 35.49 ± 0.22 | 46.12 ± 0.34 | 98.38 ± 0.7 | 99.91 ± 0.01 | 69.94 ± 0.06 | 81.88 ± 0.06 | 61.22 ± 0.16 | 77.11 ± 0.13 |
| MF | 43.62 ± 0.08 | 51.75 ± 0.14 | 95.52 ± 0.72 | 99.54 ± 0.08 | 83.29 ± 3.35 | 89.75 ± 1.9 | 29.64 ± 7.3 | 65.87 ± 8.37 |
| MLP | 35.32 ± 0.74 | 51.09 ± 0.37 | N/A | N/A | 5.36 ± 0.0 | 22.01 ± 0.01 | 61.29 ± 0.07 | 76.94 ± 0.1 |
| GCN | 43.17 ± 0.36 | 54.93 ± 0.14 | 97.65 ± 0.68 | 99.85 ± 0.06 | 81.48 ± 0.48 | 89.62 ± 0.23 | 70.77 ± 0.34 | 85.43 ± 0.18 |
| GAT | 42.07 ± 1.51 | 53.45 ± 0.64 | 98.15 ± 0.24 | 99.93 ± 0.02 | OOM | OOM | OOM | OOM |
| SAGE | 43.02 ± 0.63 | 54.38 ± 0.27 | 99.17 ± 0.11 | 99.98 ± 0.01 | 81.84 ± 0.24 | 89.46 ± 0.13 | 71.91 ± 0.1 | 85.86 ± 0.09 |
| GAE | OOM | OOM | 28.29 ± 13.65 | 48.34 ± 15.0 | OOM | OOM | OOM | OOM |
| SEAL | 43.5 ± 1.75 | 49.25 ± 1.39 | 82.42 ± 3.37 | 92.63 ± 2.05 | 87.34 ± 0.49 | 92.45 ± 0.26 | 65.11 ± 2.33 | 77.64 ± 2.43 |
| BUDDY | 50.57 ± 0.18 | 55.63 ± 0.68 | 97.81 ± 0.31 | 99.93 ± 0.01 | 82.5 ± 0.51 | 88.36 ± 0.32 | 67.47 ± 0.32 | 81.94 ± 0.26 |
| Neo-GNN | 46.93 ± 0.17 | 53.81 ± 0.19 | 83.45 ± 11.03 | 94.7 ± 4.82 | 81.21 ± 1.39 | 88.31 ± 0.19 | 62.14 ± 0.51 | 79.13 ± 0.42 |
| NCN | 45.89 ± 1.11 | 52.36 ± 0.33 | 98.43 ± 0.22 | 99.96 ± 0.01 | 89.37 ± 0.28 | 93.11 ± 0.27 | 71.56 ± 0.03 | 84.01 ± 0.05 |
| NCNC | 44.76 ± 4.64 | 52.41 ± 2.09 | >24h | >24h | 91.0 ± 0.24 | 94.72 ± 0.18 | 72.85 ± 0.9 | 86.35 ± 0.51 |
| NBFNet | OOM | OOM | >24h | >24h | OOM | OOM | OOM | OOM |
| PEG | 38.71 ± 0.17 | 49.34 ± 0.70 | 84.21 ± 9.2 | 95.76 ± 3.48 | OOM | OOM | OOM | OOM |

Table 17: Dataset Licenses.

| Datasets | License |
| --- | --- |
| Cora | NLM License |
| Citeseer | NLM License |
| Pubmed | NLM License |
| ogbl-collab | MIT License |
| ogbl-ddi | MIT License |
| ogbl-ppa | MIT License |
| ogbl-citation2 | MIT License |

efficiency of the evaluation process, especially in scenarios where a significant number of positive samples exist. Nonetheless, this limitation does not detract from the potential benefits of HeaRT in providing a more realistic and meaningful link prediction evaluation setting. Furthermore, as each evaluation node pair is independent, it offers scope for parallelization, mitigating any potential efficiency concerns to a large extent. Future work can investigate ways to optimize this process.

# L  Social Impact

Our method HeaRT harbors significant potential for positive societal impact. By aligning the evaluation setting more closely with real-world scenarios, it enhances the applicability of link prediction research. This not only contributes to the refinement of existing prediction methods but also stimulates the development of more effective link prediction methods. As link prediction has far-reaching implications across numerous domains, from social network analysis to recommendation systems and beyond, improving its performance and accuracy is of paramount societal importance. We also carefully consider the broader impact from various perspectives such as fairness, security, and harm to people. No apparent risk is related to our work.