# OpenReview forum: "Evaluating Graph Neural Networks for Link Prediction: Current Pitfalls and New Benchmarking"
_NeurIPS.cc/2023/Track/Datasets_and_Benchmarks — NeurIPS 2023 Datasets and Benchmarks Poster_

### Official Review · Reviewer_KDed · 2023-07-05

**Rating:** 8
**Confidence:** 4
**Correctness:** Yes.
**Clarity:** Yes.

**Strengths:**

1. The paper unveal an important but unfounded positive and negative edge issue on the evaluation of link prediction problem
2. The paper propose a new evaluation setting more aligning with the real-world scenario.
3. Comprehensive experiments indicate the new potential provided in the link prediction domain.

**Additional Feedback:**

1. Some details on heuristic metghods seems missed. What is the maximum path length for shortest path and Katz?

**Documentation:**

Yes.

**Ethics:**

No.

**Limitations:**

Yes.

**Opportunities For Improvement:**

The Observation 1: Better than Reported Performance is of practical value since those GNN forlink prediction are generally compliciated and hard to deploy in the real-world scenario. Despite the better reported performance on the dataset. I think it could be better if you can provide a comparison of the time complexity. It could help the practitionaer better select the models with tradeoff between time complexity and the model performance.

**Relation To Prior Work:**

Yes.

**Summary And Contributions:**

1. The paper conducts a fair and reproducible comparison of different link prediction models across multiple datasets.
2.  The paper identifies shortcomings in the current evaluation setting for link prediction, e.g., unrealistic and easy negative samples. To address these issues, the paper proposes a new evaluation strategy called Heuristic Related Sampling Technique.
3. The paper presents the results of the fair comparison under the existing evaluation setting and discusses the main observations.

---

> ### Author Response · Authors · 2023-08-13
> **Response to Reviewer KDed**
>
> ### Thank you for your reviewing our paper. Please see our response to each of your questions.
>
> > Q1:   I think it could be better if you can provide a comparison of the time complexity. It could help the practitionaer better select the models with tradeoff between time complexity and the model performance.
>
> Thanks for the suggestion. We present the running time for a single seed on different model/dataset in the following table.  It shows  the GPU memory (G=Gigabyte) required and the time (h = Hour) needed to finish training with a single seed. OOM means it's requires over 50Gb of GPU memory. N/A means not applicable. It's because the ogbl-ddi doesn't have node feature and we cannot apply MLP on it. '>24h/epoch' means it needs over 24 hours to finish running one epoch.
>
> ||Cora  |Citeseer|	Pubmed|	ogbl-collab| ogbl-ddi|	ogbl-ppa|	ogbl-citation2 |
> |:------:|:------:|:-------:|:-------:|:-------:|:-------:|:-------:|:-------:|
> |Node2Vec|	(32G, 1h)|	(32G, 1h)	|(32G, 1h)	|(32G, 4h)	|(32G, 4h)	|(32G, 4h)|	(32G, 6h)
> |MLP|	(32G, 1h)|	(32G, 1h)	|(32G, 1h)	|(32G, 4h)|	N/A	|(32G, 4h)|	(32G, 6h)
> |MF|	(32G, 1h)|	(32G, 1h)	|(32G, 1h)	|(32G, 4h)	|(32G, 4h)	|(32G, 4h)|	(32G, 6h)
> |GCN|	 (32G, 1h)|	(32G, 1h)|	(32G, 1h)	|(32G, 4h)|	(32G, 4h)|	**(32G, 24h)**|	(32G, 6h)
> |GAT|	 (32G, 1h)|	(32G, 1h)|	(32G, 1h)	|(32G, 4h)|	(32G, 4h)|OOM|	OOM
> |SAGE|	 (32G, 1h)|	(32G, 1h)|	(32G, 1h)	|(32G, 4h)|	(32G, 4h)|	**(32G, 24h)**|	(32G, 6h)
> |GAE|	 (32G, 1h)|	(32G, 1h)|	(32G, 1h)	|(32G, 4h)|	(32G, 4h)|	OOM|OOM
> |SEAL|	(32G, 1h)|	(32G, 1h)	| **(32G, 24h)**|	(32G, 4h)|	**(32G, 12h)**	|**(32G, 48h)**|	**(32G, 48h)**
> |BUDDY|	(32G, 1h)	|(32G, 1h)	|(32G, 1h)	|(32G, 1h)	|(32G, 1h)	|(32G, 4h)|	(32G, 10h)
> |NeoGNN	|(32G, 1h)|	(32G, 1h)|	(32G, 1h)|	(32G, 4h)|	(32G, 4h)|	(32G, 6h)|	**(48G, 10h)**
> |NCN|	(32G, 1h)|	(32G, 1h)|	(32G, 1h)|	(32G, 1h)|	(32G, 1h)|	(32G, 10h)	|**(48G, 4h)**
> |NCNC|	(32G, 1h)|	(32G, 1h)|	(32G, 1h)|	**(48G, 1h)**|	**(48G, 48h)**|	**(48G, 48h)**|	**(48G, 24h)**
> |NBFNet|	(32G, 1h)|	(32G, 1h) |	**(32G, 20h)**	|OOM|	>24h/epoch|	OOM|	OOM
> |PEG|	(32G, 1h)|	(32G, 1h)|	(32G, 1h)|	(32G, 1h)|	(32G, 1h)|	OOM	|OOM
>
> > Q2: What is the maximum path length for shortest path and Katz?
>
> We don't set a maximum path length when computing the shortest path.
>
> For Katz, our computation differs by dataset. For smaller dataset, we don't use a maximum path length. Rather we use the closed-form solution for calculating Katz. However, calculating the closed-form solution is impractical on the larger datasets. For those datasets, we instead calculate Katz iteratively using a maximum path length of $3$. The downside to this is that it can still be very costly to run. As shown in Tables 2 and 5 in our paper, Katz typically requires over 24 hours to compute for most of the OGB datasets.
>
> We recently re-ran Katz and found that obtaining results with a maximum length setting of 2 is feasible. To enhance the comprehensiveness of our study, we've updated the paper accordingly in Table 2, Table 5, Table 13, and Table 15. We will configure the maximum length to be either 3 or 2, depending on the feasibility of obtaining the results within 24 hours.

---

### Official Review · Reviewer_dRZC · 2023-07-09
**Evaluating Graph Neural Networks for Link Prediction: Current Pitfalls and New Benchmarking**

**Rating:** 7
**Confidence:** 4
**Correctness:** Yes.
**Clarity:** Yes.

**Strengths:**

1. The paper finds several important pitfalls in the link prediction tasks with suitable solution
2. The proposed evaluation setting is more aligned with the real-world scenario.
3. Comprehensive experiments indicate the new potential provided in the link prediction domain.

**Additional Feedback:**

1. what does NA stands for in the result tables.

**Documentation:**


Yes.

**Ethics:**

No.

**Limitations:**

Yes.

**Opportunities For Improvement:**

The preliminary study in Figure 1 indicate the pitfall on the current evaluation setting. Nonetheless, there is no corresponding experiment on the new proposed evaluation setting. Despite the algorithm aims to select hard negative edges with more common neighbor. I would suggest there is comprehensive experimental results to support this claim.

**Relation To Prior Work:**

Yes

**Summary And Contributions:**

1.  The paper conducts a fair comparison of different link prediction models on multiple datasets. All models are tuned using the same hyperparameters, and various evaluation metrics are employed.
2. The paper emphasizes the importance of aligning the evaluation setting with real-world scenarios. The existing evaluation procedure fails to consider the specific target node and uses randomly selected negative samples. By proposing a new evaluation setting that incorporates relevant negatives tailored to each positive sample, the paper aims to improve the relevance and applicability of link prediction methods in real-world contexts.
3. The paper presents a detailed performance analysis of the evaluated link prediction methods under the existing evaluation setting ad new setting. It discusses key observations regarding the underreported performance of vanilla due to poor hyperparameter tuning.

---

> ### Author Response · Authors · 2023-08-13
> **Response to Reviewer dRZC**
>
> ### Thank you for your reviewing our paper. Please see our response to each of your questions.
>
> > Q1:  Comparison with figure 1 under HeaRT
>
> We appreciate the suggestion. We include a comparison of the CN distribution for the original and HeaRT negatives in Figures 4 and 5 in Appendix A. We further include the CN distribution for the positive samples as a comparison. We observe that for all datasets, the distribution of the negatives produced by HeaRT is much closer to the positive distribution than the original setting.
>
>
> > Q2:   what does NA stands for in the result tables.
>
> N/A means it is not applicable. This only applies to when we are training a MLP for ogbl-ddi. Since ogbl-ddi has no node features, we are unable to train a MLP. We revised our paper on line 166 to make this clearer.

---

> > ### Comment · Reviewer_dRZC · 2023-08-18
> >
> > Thank you for your reply. I have no more questions.

---

### Official Review · Reviewer_P7Ue · 2023-07-18
**Introducing a new negative sampling is good, but inappropriate to make benchmarking existing methods using it.**

**Rating:** 6
**Confidence:** 4
**Clarity:** yes.

**Strengths:**

S1. The study conducts various experiments using seven representative datasets and 19 existing techniques. Through experiments, it reveals the shortcomings of existing benchmarks, including cases where higher accuracy is achieved than reported in the original papers, large standard deviations in accuracy, inconsistencies between evaluation metrics such as AUC and Hits@K, and the issues with using accuracy reported in the original papers.

S2. Evaluating using multiple metrics is valuable. Particularly, Hits@K is a more appropriate evaluation metric, since AUC does not provide the metric from users' perspective.

S3. Categorizing existing techniques into four types (Heuristic, embedding, GNN, GNN+pairwise info) is appropriate and clear.

S4. Through the benchmarking experiments using the proposed negative sampling technique, this study gives new insights on existing methods. Notably, some heuristic methods outperform the latest techniques such as GNN in ogbl-collab. Additionally, a significant reduction in the standard deviation of accuracy is achieved.

**Additional Feedback:**

Not anymore.

**Correctness:**

It is inappropriate to make benchmark existing methods by changing their pretraining setting, the ratio between positive/negative samples, and their negative sampling technique.

**Documentation:**

The reproducibility is good.

**Ethics:**

No ethical concerns.

**Limitations:**

L1. It is inappropriate to make benchmark existing methods by changing their pretraining setting, the ratio between positive/negative samples, and their negative sampling technique, because those existing methods are designed based on those pretraining settings, the positive/negative sampling ratio, and the negative sampling technique.
In this sense, it is reasonable that the accuracies for BUDDY and Neo-GNN decrease, if we change their pretraining setting and the positive/negative sampling ratio.

L2. Efficiency in the test phase is important in practice. In that sense, the proposal (HeaRT) may not be practical as the paper described, "the large size of the validation and test sets (see Table 7 in Appendix B) makes HeaRT prohibitively expensive".

**Opportunities For Improvement:**

O1. The idea of using hard negative samples has already been discussed in the paper "Understanding Negative Sampling in Graph Representation Learning, KDD 2020." A comparative discussion between this paper and HeaRT should be made.

O2. Using hard negative samples (samples near the class decision boundary) should be a correct approach for learning more accurate class boundaries. However, since the accuracy significantly is decreased as shown in Table 1 (existing evaluation setting) and Table 4 (new setting using HeaRT), it seems that using hard negative samples does not help increase the model quality. Here, I like to confirm that the test dataset is the same between the existing evaluation setting and the new setting using HeaRT. If they are the same, the authors should provide a sufficient explanation for the decrease in model accuracy due to using HeaRT.

O3. The meaning of "corruption of positive sample" is not clear, which makes it difficult to understand the details of the proposed technique. A more detailed explanation should be added.

O4. While it is intriguing that the heuristic method Katz achieves the highest performance in ogbl-collab, a more detailed analysis should be included in the paper, such as what types of graph topologies are effectively captured by Katz but not by GNNs.

O5. The necessity of using a unified data split across multiple datasets is unclear. Using standard splits for each dataset should be sufficient. A more detailed explanation should be added.

**Relation To Prior Work:**

This paper lacks important related work [1]. A comparative discussion should be provided.
[1] "Understanding Negative Sampling in Graph Representation Learning, KDD 2020."

**Summary And Contributions:**

This research focuses on the problem of benchmarking for link prediction.
The main contributions are as follows.
1) This research identifies several important issues in existing benchmarks (See S1 in the Strengths).
2) It introduces a new negative sampling technique to solve the above issues and then gives new insights on existing methods by conducting various experiments using seven representative datasets and 19 existing techniques (See S4 in the Strengths).

---

> ### Author Response · Authors · 2023-08-13
> **Response to Reviewer P7Ue [Part 1/2]**
>
> ### Thank you for helping us improve our paper. We respond to all your comments and suggestions below.
> > Q1: A comparative discussion between ‘Understanding Negative Sampling in Graph Representation Learning, KDD 2020.’ and HeaRT should be made.
>
> We respectively point out that **the shared paper is different from our work**. In our work, we are concerned with how negative samples effect **evaluation** while they are concerned with their effect **during training**.
> Specifically, in their paper, they study the influence of negative sampling on the training optimization objective. They then propose a method to efficiently generate such samples during training via Markov chain sampling. In contrast, HeaRT  is **exclusively applied during the evaluation process**, leaving the training stage unaffected. This critical distinction delineates a fundamental difference in both application and focus between the two approaches.  **While they focus on how to use negative sampling to better train methods, we focus on how to use them to better evaluate methods.**
> Given these underlying differences, a direct comparison between this paper and the HeaRT method may not be appropriate.  We included a small discussion on lines 277-280 in the revision to make this distinction clearer.
>
>
> > Q2: Using hard negative samples (samples near the class decision boundary) should be a correct approach for learning more accurate class boundaries. It should improve the accuracy.
>
> As noted in the last comment, our proposed HeaRT setting is only used  **during evaluation**.  More specifically, the test negative samples in the HeaRT are **different** from those in the existing setting (the test positive samples are the same). Thus the test data is **different** between these two settings, and the training procedure is the same as the existing setting. As such, the HeaRT setting does not directly influence the model's decision boundary. By introducing hard negative samples during evaluation, HeaRT inherently makes the task more challenging. The subsequent decrease in performance observed under the HeaRT setting is an expected outcome, reflecting the increased difficulty in evaluation.
>
> > Q3: The meaning of "corruption of positive sample" is not clear, which makes it difficult to understand the details of the proposed technique. A more detailed explanation should be added.
>
> We apologize for not making this clearer in our paper. For a positive sample $(u, a)$, the corruption replaces only one of the two nodes in the pair. E.g., $(u, v)$ or $(v, a)$. This concept is formally defined in Equation (3) in our paper:
> \begin{equation}
>     S(u, a) = \lbrace(u', a)  | u' \in \mathcal{V}\rbrace \cup \lbrace (u, a')  | a' \in \mathcal{V}\rbrace,
> \end{equation}
> where
> $S(u,a)$ is the set with all possible corruptions of the positive sample  $(u,a)$. We updated line 77 in the revision to make it clearer.
>
> > Q4: Katz achieves the highest performance in ogbl-collab, a more detailed analysis should be included in the paper, such as what types of graph topologies are effectively captured by Katz but not by GNNs.
>
> We further analyzed the performance of Katz on the ogbl-collab dataset. Interestingly, we find that it's due to the dataset being a dynamic graph. In this dataset, nodes represent authors and edges represent a collaboration between two authors. Each edge further includes an edge attribute that specifies the year of collaboration. For example, each edge takes the form (Author 1, Year, Author 2). The task is to predict collaborations in 2019 (test) based on those until 2017 (training) and 2018 (validation).
>
> We found that of the positive samples in the test set, around 46% also appear as positive samples in the training set. For example, the edge (Author 1, 2017, Author 2) may be in the training data while (Author 1, 2019, Author 2) may also appear in the test data. This is because two authors who collaborated in the past are apt to do so again in the future. As such, when evaluating the test sample (Author 1, 2019, Author 2), they will have a path of length 1 between them. Since Katz considers paths of shorter length as more important, the inclusion of these length 1 paths has a large effect on the score. Furthermore, since this phenomena is common among positive samples but rare among negatives, it allows for an easy way to classify many test links.
>
> We note that since all other models keep these ``repeated'' samples in the training graph, we also keep them when computing the Katz. However, we found that under the existing setting we accidentally omitted these links when computing Katz. The updated results on ogbl-collab under the existing setting is 64.33% (Note: We further updated the results for Shortest Path).
>
> We appreciate the reviewers for bringing this aspect to our attention. We believe it will enhance the clarity and depth of our paper. We have updated the results in revision and added these observations to the Appendix (see Appendix H).

---

> > ### Author Response · Authors · 2023-08-13
> > **Response to Reviewer P7Ue [Part 2/2]**
> >
> > > Q5: The necessity of using a unified data split across multiple datasets is unclear. Using standard splits for each dataset should be sufficient.
> >
> > We apologize for the misunderstanding. By ``unified data split across multiple datasets", **we mean using a standard split for each dataset individually**. Previously, for Cora, Citeseer, and Pubmed multiple splits existed in literature for each specific dataset. For example, for Cora some studies used a fixed split of 85/5/10 for the train/valid/test while others used 10 random splits of 70/10/20. Thus it's difficult to compare those works because different splits are adopted. Therefore, one of the goals in our study was that for each dataset, we evaluate all methods under the same splits. As such, for Cora, Citeseer, and Pubmed **we use a standard split for each individually** when training and evaluating different methods.
> >
> > We apologize for the confusion. We've updated lines 40-42 and 47-48 in the revision
> > to make this distinction clearer.
> >
> > > Q6: It is inappropriate to make benchmark existing methods by changing their pre-training setting, the ratio between positive/negative sample, and their negative sampling technique,
> >
> > Only a few modifications were made to those components that were not integral parts of each framework and **are not mentioned in their original papers**. Rather, they were discovered when examining their code. These changes **only apply to the ogbl-ddi dataset** and are listed in Appendix C.2. They include:
> >
> >  * We don't use any pre-training for Neo-GNN and NCNC.
> > * For BUDDY, we use the same number of positive and negative samples when training.
> >
> > Furthermore, we note that the modifications made were motivated by the fundamental principle of ensuring a fair comparison among all the models, which is crucial for rigorous benchmarking. Since these are very general strategies that aren't unique to each method, they serve as an unfair advantage to those methods that use them. Nevertheless, since we only made changes to three methods in one particular dataset to ensure a fair comparison, the general observations will not be affected.
> >
> > > Q7:  Efficiency in the test phase is important in practice. In that sense, the proposal (HeaRT) may not be practical as the paper described, "the large size of the validation and test sets (see Table 7 in Appendix B) makes HeaRT prohibitively expensive".
> >
> > We acknowledge that we might introduce a large amount of negative samples when the number of positive samples is very large.
> > However, as the evaluation for each node pair is independent, it offers scope for parallelization, mitigating potential efficiency concerns. Also, since evaluation is done  **offline** with collected data, it only needs to be done once. As such, HeaRT doesn't compromise the complexity of the model itself. Therefore when the models are launched for real-world applications, the efficiency of **online** inference remains unaffected by the HeaRT evaluation process, as HeaRT is only applied **offline**.
> >
> > Lastly, the evaluation in HeaRT is more realistic and aligns well with the online inference of real-world applications.
> > More specifically, the HeaRT setting customizes a set of negatives to each positive sample, selecting those that are more relevant to the corresponding positive samples. This offers a more accurate alignment between offline evaluation and real-world applications compared to existing methods, which often rely on the random  selection of negative samples.

---

> > > ### Comment · Reviewer_P7Ue · 2023-08-14
> > >
> > > Q5, Q6: Good. I agree with the authors.
> > >
> > > Q7: I understand that there is a trade-off between efficiency and accuracy in benchmarking.
> > >
> > > As a whole, I raised the score from 4 to 6.

---

> > > > ### Author Response · Authors · 2023-08-14
> > > >
> > > > We thank the reviewer for raising their score.
> > > >
> > > > Please let us know if you have any more questions.

---

> > ### Comment · Reviewer_P7Ue · 2023-08-14
> >
> > Q1 and Q2: I understand the difference between HeaRT (during testing) and KDD 2020 (during training).
> > Q3: The revision is good.
> > Q4: Good further analysis.

---

### Official Review · Reviewer_rGWc · 2023-07-22

**Rating:** 6
**Confidence:** 4
**Correctness:** Yes.
**Clarity:** Yes.

**Strengths:**

1. The writing is overall clear. The overall organization is satisfying.
2. The introduced observations and the solutions are reasonable.
3. The experimental results are comprehensive and can basically reflect the claims in the paper.

**Additional Feedback:**

The response to the second concern will be used to decide whether this paper is possible to be accepted by top conferences like NeurIPS, while the first and third concern may prevent this paper to be accepted in the current submission.

**Documentation:**

No.

**Ethics:**

No.

**Limitations:**

Yes.

**Opportunities For Improvement:**

1. The preliminary is unclear. The prediction tasks and the evaluation pitfalls lack formal formulations.
2. The rationality behind the heuristic sampling is not clear. Specifically, it is not convincing that including these heuristic sampling strategies can evaluate methods **fairer** instead of **introducing extra inductive biases** that might facilitate some methods, leading to unfair comparisons.
3. The code quality is not satisfying:
* Lack of dependency installation explanation. requirements.txt is not enough. Considering the use of torch and GPUs, it is necessary to state proper system and driver requirements. A better practice is to automate the whole process.
* Lack of documentation of classes and functions.
* Repeating and unused code. Please DRY (don't repeat yourself).

Please notice that for a benchmark paper, the code quality should be at least satisfying.

**Relation To Prior Work:**

Yes.

**Summary And Contributions:**

The paper discusses the pitfalls of current evaluations of link prediction tasks and proposes a novel strategy to evaluate. Specifically, instead of using a simple and global negative sampling strategy, the authors apply several heuristic strategies to filter negative candidates. The experiment results are comprehensive and self-contained.

---

> ### Author Response · Authors · 2023-08-13
> **Response to Reviewer rGWc [Part 1/3]**
>
> ### Thank you for reviewing. We respond to your comments below.
> > Q1: The prediction tasks and the evaluation pitfalls lack formal formulations.
>
> We updated our paper with more formal definitions. Please see Section 2.1 for a formal definition of link prediction, Appendix B.1 for a definition of the various evaluation metrics, and Appendix B.2 for a definition of the negative sampling procedure. These changes are in red to indicate they are new. We further provide the formulations below for ease of access:
>
>  > **Task formulation**
>   We formally define the task of link prediction. Let $\mathcal{G} = \{ \mathcal{V}, \mathcal{E} \}$ be a graph where $\mathcal{V}$ and $\mathcal{E}$ are the set of nodes and edges in the graph, respectively. Furthermore, let $X \in \mathbb{R}^{ |\mathcal{V}| \times d}$ be a set of $d$-dimensional features for each node. Link prediction aims to predict the likelihood of a link existing between two nodes given the structural and feature information. For a pair of nodes $u$ and $v$, the probability of a link existing, $p(u, v)$, is therefore given by: \begin{equation}p(u, v) = p(u, v  |  \mathcal{G}, X)\end{equation}.
>  Traditionally, $p(u, v)$ was estimated via non-learnable heuristic methods. More recently, methods that use learnable parameters have gained popularity. These methods attempt to estimate $p(u, v)$ via a learnable function $f$ such that:
> \begin{equation}
>     p(u, v) = f(u, v  |  \mathcal{G}, X, \Theta),
> \end{equation}
> where $\Theta$ represents a set of learnable parameters. A common choice of $f$ is graph neural networks.
>
> > **Training Negative Samples**. During training, the negative samples are randomly selected, with all nodes being equally likely to be selected. Let $\mathcal{V}$ and $\mathcal{E}$ be the set of nodes and edges in $\mathcal{G}$. Furthermore, we define $v \in \text{Rand}(\mathcal{V})$ as returning a random node in $\mathcal{V}$. A single negative sample $(a^{-}, b^{-})$ is given by:
> \begin{equation}
>     (a^{-}, b^{-}) = \left(\text{Rand}(\mathcal{V}), \text{Rand}(\mathcal{V}) \right).
> \end{equation}
> Typically one negative sample is generated per positive sample.
>
> > **Evaluation Negative Samples**. For the existing setting, a fixed set of randomly selected samples are used as negatives during evaluation. Furthermore, the same set of negative samples are used for each positive sample. This is equivalent to $(a^{-}, b^{-})$. The only exception is ogbl-citation2, where each positive sample is only evaluated against its own set 1000 negative samples. For a positive sample, its negative samples are restricted to contain one of its two nodes (i.e., a corruption). The other node is randomly selected from $\mathcal{V}$. This is equivalent to selecting a set of random samples from the set $S(a, b)$,
> \begin{equation}
>     S(a, b) =  \lbrace (u', b)  |  u' \in \mathcal{V}\rbrace \cup \lbrace u', a) |  u' \in \mathcal{V} \rbrace .
> \end{equation}
>
> > **Evaluation Metrics**
> We define the various evaluation metrics used. Given a single positive sample and $M$ negative samples, we first score each sample and then rank the positive sample among the negatives. The rank is then given by $\text{rank}_i$, i.e., a rank of 1 indicates that the positive sample has a higher score than all negatives. The hope is that the positive sample ranks above most or all negative samples. We use $N$ to denote the number of positive samples.
>  > * **Hits@K**. It measures whether the true positive is within the top K predictions or not:   $\text{Hits@K} = \frac{1}{N}\sum_{i=1}^{N}\mathbf{1}(\text{rank}_{i} \leq \text{K})$. $\text{rank}_i$ is the rank of the $i$-th sample. The indicator function  **1** is 1 if $\text{rank}_i \leq \text{K}$, and 0 otherwise.
> > *  **Mean Reciprocal Rank (MRR)**. It is the mean of the reciprocal rank over all positive samples: $\text{MRR}=\frac{1}{N}\sum_{i=1}^{N}\frac{1}{\text{rank}_i}$, where $\text{rank}_i$ is the rank of the $i$-th sample.
> > * **AUC**. It measures the likelihood that a
>  positive sample is ranked higher than a random negative sample: $\text{AUC} = \frac{\sum_{i\in \mathcal{D}^0} \sum_{j\in \mathcal{D}^1} \mathbf{1}(\text{rank}_i < \text{rank}_j)}{|\mathcal{D}^0|\cdot |\mathcal{D}^1|}$, where $\mathcal{D}^0$ is the set of positive samples, $\mathcal{D}^1$ is the set of negative samples, and $\text{rank}_i$ is the rank of the $i$-th sample. The indicator function **1** is 1 if $\text{rank}_i < \text{rank}_j$, and 0 otherwise.
>
>  Lastly, we note that the first two evaluation pitfalls, *1) Lower than Actual Performance* and *2) Lack of Unified Settings on the Planetoid Datasets*, are observations about the performance and data split and are therefore not accompanied by any detailed definitions. For the 3rd pitfall, *3) Unrealistic Evaluation Setting*, we formally define the current negative sampling procedure (see definition above). Furthermore, the new evaluation procedure is formulated in Section 4.2, with additional details in Appendix G.

---

> > ### Author Response · Authors · 2023-08-13
> > **Response to Reviewer rGWc [Part 2/3]**
> >
> > > Q2: Specifically, it is not convincing that including these heuristic sampling strategies can evaluate methods  **fairer instead of introducing extra inductive biases** that might facilitate some methods, leading to unfair comparisons.
> >
> > We respectively point out that our new evaluation setting is not related to the inductive bias. This is because inductive biases can only be introduced in training and algorithmic design, **while our focus is only on the evaluation (or test) stage**. As a reminder,   *inductive biases* are assumptions made regarding the search space of possible solutions. The inductive bias is often reflected through the learning algorithm itself, regularization terms (e.g., L1 assumes a sparse solution), or other mechanisms (please see "Relational inductive biases, deep learning, and graph networks (2018)" for a good discussion). Therefore because the HeaRT setting is applied solely during the evaluation, it does not effect the inductive biases of any specific model.
> >
> > We also note that our main motivation in creating HeaRT is to introduce a more realistic evaluation setting for link prediction. This is due to two issues with the existing setting:
> >  1) In a real-world scenario, we typically care about predicting links for a specific node. For example, in friend recommendations, we aim to recommend friends for a specific user $u$. To evaluate such models for $u$, we strive to rank node pairs including $u$.  **However, under the current setting all positive samples are compared against the same set of negative samples**. This situation does not reflect how real world link prediction works.
> >
> > 2) The negative samples used are too easy. As shown in Figure 1 in our paper, almost all of them have 0 CNs (a strong heuristic). This is a function of how the negatives are generated. Since they are randomly selected, many are irrelevant and trivially non-existent. **However in real-world applications, candidates links are selected via a candidate generation algorithm that produce a set of relevant candidates. Heuristics are often used for this task**. See [39, 40] in the revision for a discussion of candidate generation at Twitter and Pinterest.
> >
> >
> > We address the 1st limitation by personalizing the set of negative to each positive sample. Specifically, for a positive sample $(a, b)$ we want any candidate links (i.e., negatives) to contain one of $a$ or $b$. As such, the pool of potential negatives for $(a, b)$ is given by the following where $\mathcal{V}$ is the set of nodes:
> > \begin{equation}
> >     S(a, b) = \lbrace (u', b)  |  u' \in \mathcal{V}\rbrace \cup \lbrace(u', a)  |  u' \in \mathcal{V}\rbrace.
> > \end{equation}
> >
> > We address the 2nd limitation by designing our own candidate generation algorithm to select samples from $S(a, b)$. We do so by combining multiple commonly-used heuristics that incorporate different types of information (i.e., local, global, and feature). For example, both CNs and PPR (used by us) are shown in [39] by Twitter.
> >
> > Interestingly, we observe that  **due to the unrealistic nature of the existing evaluation setting, it can sometimes promote an unfair comparison between methods**. For example, under the existing setting the vast majority of negative samples have no CNs (see Figures 1 and 3 in our paper). This provides an advantage to methods that use the CN information in their model, as they can easily differentiate between positive and negative samples via the CNs.
> >
> > However,  **we don't observe this problem under HeaRT**. As shown in Figures 4 and 5 in Appendix A, the distribution of CNs for the HeaRT negative samples is much more similar to that of the positive samples. This provides less of a distinction between positive and negative samples, allowing for a fairer comparison between methods.  Furthermore, by utilizing a mix of different types of information to generate negative samples, HeaRT is more robust to such comparison issues.

---

> > > ### Author Response · Authors · 2023-08-13
> > > **Response to Reviewer rGWc [Part 3/3]**
> > >
> > > > Q3: Code quality is not satisfying.
> > >
> > > Thank you for your feedback on our code quality. We hope to make our code as accessible and easy to use as possible. We address several of your concerns below.
> > >
> > > 1. We included a detailed installation guide in our Github repo (see [here](https://github.com/Juanhui28/HeaRT/blob/master/installation.md) for mode details). This includes all steps to install our package, how to obtain the data, and the various requirements. This includes the GPU driver, system, and python versions.
> > >
> > > 2. We provide extensive documentation on how to use our code in the [Readme](https://github.com/Juanhui28/HeaRT/blob/master/README.md). We first demonstrate how to reproduce our results. For each setting and dataset, we include the command to run all possible methods. These commands include the optimal hyperparameters. For example, the file `scripts/hyperparameters/existing_setting_ogb/ogbl-collab.sh` contains the command for running each method under the existing setting for the ogbl-collab dataset. We also detail how to generate the negative samples used by HeaRT. To allow easy replication, we further include the commands to run this script for each dataset. They are placed in the `scripts/HeaRT` directory. For example, the `generate_cora.sh` script produces the negatives used by HeaRT for Cora. We further explain how to modify these scripts to produce your own custom set of negative samples.
> > >
> > > 3. We agree that proper code documentation and removing unused/repeated code are vital to a well-functioning codebase. Because of this we've been working on a new version of our repository since after the submission (See [here](https://github.com/Juanhui28/HeaRT/tree/version2) for more details. Note that we've been working on the high-level design and haven't implemented most methods yet.). Our new version removes the duplicate code and provides a shared codebase upon which the different methods are implemented. We are also documenting the different components of our codebase to ensure ease of use. We hope to have a version ready for public use in the near future.

---

> ### Author Response · Authors · 2023-08-17
>
> Thank you again for reviewing our paper. Your comments have been very helpful in improving our paper.
>
> We are writing to check if you have any further concerns or questions after reading our comments. Since attending to any additional comments may require extensive experiments or revisions, we want to ensure that we have sufficient time to address any concerns the reviewers may have.
>
> If you do have any additional concerns or questions, please let us know. We’d be happy to try and address them.
>
> Thank you!

---

> > ### Comment · Reviewer_rGWc · 2023-08-17
> >
> > Thank you for your response. I appreciate your tremendous efforts during this period.
> >
> > My first concern has been addressed satisfactorily. Your response to my second concern is reasonable, so I agree that contemporarily there is no better way to achieve a fairer evaluation than your work. After reevaluating your code, I think your first and second points are satisfactory, but the third one is not convincing since the new version hasn't been released. However, according to the efforts you made, I would give credit to your future updates.
> >
> > Update: $4\rightarrow 6$
> >
> > Please continue updating your repo accordingly.

---

> > > ### Author Response · Authors · 2023-08-17
> > >
> > > We thank the reviewer for raising their score.
> > >
> > > We plan on continuing to work on our new version and will release it as soon as it is ready.
> > >
> > > Please let us know if you have any more questions.

---

### Official Review · Reviewer_vjFt · 2023-07-23
**A very useful and thorough contribution, but not sure if it is substantial enough for NeurIPS**

**Rating:** 6
**Confidence:** 3
**Correctness:** I believe all claims are correct and …

**Strengths:**

Review experiments* like these are always useful and powerful contributions and should be lauded as a key contribution to the field. (Ideally, every subfield of machine learning should regularly re-test its state-of-the-art models).

Making the evaluation more realistic and challenging is also a sound contribution. This is clear from the fact that KG models are already evaluated in a way similar to what the authors propose.

* That is, experiments, which introduce no new models, but take a set of published models and redo their training under uniform circumstances for a fair comparison.

**Additional Feedback:**

Some minor points to consider in the next revision:

- Citation [42] seems to have been included in error.
- I'm not sure where the a in HeaRT comes from.
- The authors seem _very_ fond of the word "utilize". It may sound fancy, but it just means "use". I suggest sticking to the latter.
- The same goes for "leverage".
- A period goes before a footnote mark.
- "pitfalls that befall recent work" is a mixed metaphor.

**Clarity:**

The paper is clearly written and easy to read, save for a few stylistic issues detailed below.

**Documentation:**

The code seems well-documented. See above for my point on publishing and documenting the data as well.

**Ethics:**

No concerns.

**Limitations:**

See above.

**Opportunities For Improvement:**

My reasons for only recommending a marginal accept are two-fold.

Firstly, the spread of the hyperparameters selected seems limited. A useful point of comparison here is [8], similar work done and accepted at a similar conference. There, the authors tested, for instance, a wider range of embedding methods, a variety of loss functions, and additions like learning rate scheduling.

It may be, that since the models used here are GNNs rather than simple embedding models, the resources are actually similar, but in that case the authors should justify their choice. That is, take a rough estimate of the resources used in previous similar studies, and take this as a rough benchmark for their own.

Second, the set of heuristics chosen for the HeaRT selection of negatives seems rather ad-hoc. The use of corruptions of the true triple rather than uniformly sampled negatives is well justified by a realistic example based on a common real-world use case. However, with the HeaRT selection, this justification is less clear. Would a different set of heuristics lead to the same sort of ranking among models, or do the results depend on the choice of heuristics? If so, how is the current selection justified?

Beyond that, it is not clear to me that the authors publish _their selected negatives_. Since this evaluation is don on the same set of negatives for all models, I would strongly recommend that the authors make this the canonical set of negatives for all datasets that should be used by all future authors claiming to evaluate with HeaRT. They should make this protocol very clear in the paper and on the github repository, and, most importantly, publish text-based versions of all datasets that are easy to read in a platform independent way. Ideally, these version of the datasets would go up on Zenodo and be linked to by DOI.

**Relation To Prior Work:**

As far as I know all related work is mentioned (but this work is a little outside of my area of expertise).

**Summary And Contributions:**

The authors provide a review experiment on link prediction in simple graphs, showing (as similar reviews in other domains have done) that with equal hyperparameter tuning, old methods often outperform newer ones.

In addition, the authors make the case that the default evaluation is likely too simple, and a more challenging evaluation is called for. They introduce such an evaluation method by:
1. Evaluating a mthod on how it ranks a true triple _among corruptions of that same triple_.
2. Using simple heuristics to pre-select a set of challenging negatives.

---

> ### Author Response · Authors · 2023-08-12
> **Response of Raised Questions [Part 1/3]**
>
> ### Thanks for your valuable comments and suggestions. We respond to your concerns below.
>
> > Q1: The hyperparameters selected seems limited. Comparison with [8]. The authors should justify their choice.
>
> Since we were evaluating a large number of methods on many datasets, we were limited in the scope of hyperparameters that we could tune. Specifically, we consider 7 datasets where 14 of the methods include tunable hyperparameters. Assuming we consider $k$ sets of hyperparameters for each method, this would result in a total of $7 * 14 * k$ different combinations. Even for a modest $k$ this can easily be prohibitively expensive.
>
> Because of the large number of datasets and models in our study, it's difficult to make any comparison between our study and [8].  Comparing with [8], they only use 5 models and 2 datasets. This allows for a much wider range of hyperparameters to be considered. Also, the datasets used in [8] tend to be much smaller. For example, the two datasets used in their work, FB15K-237 and WN18RR, have 270K and 85K edges, respectively. This is considerably lower than the smallest OGB dataset, ogbl-collab, which has 1.25M edges. Furthermore, both ogbl-ppa and ogbl-citation2 have 30M edges, necessitating much more resources. Lastly, we note that since [8] focuses on KGC and ours on link prediction we share no overlap in datasets nor methods.
>
> We further expound on the resources needed in our study.
> We demonstrate this by analyzing the memory and runtime requirements of most of the different methods. These are presented in the table below. It shows  the GPU memory (G=Gigabyte) required and the time (h = Hour) needed to finish training with **a single seed**. OOM means it's requires over 50Gb of GPU memory.  N/A means not applicable as the ogbl-ddi doesn't have node feature and we cannot apply MLP on it. '>24h/epoch' means it needs over 24 hours to finish running one epoch.
>
> From the table, we observe that certain methods like NCNC and datasets like ogbl-citation2 often require 48G of memory. Others require more time. For example, SEAL and NCNC require around 48 hours to train one seed on ogbl-ppa. **As such, tuning these models on a large search space is impractical. To address this issue and ensure a fair benchmark, certain hyperparameters were fixed at commonly accepted values (these values were chosen by inspecting the hyperparameter values of other models).**
>
> Lastly, the different methods were chosen based on their performance and popularity. Due to our limited resources, we had to select those that were most promising. For the training settings, we follow the common settings used in other works. This includes, the binary cross-entropy loss, the Adam optimizer, and using an equal amount of positive and negative samples during training. We also don't apply pre-training to any model.
>
>
> ||Cora  |Citeseer|	Pubmed|	ogbl-collab| ogbl-ddi|	ogbl-ppa|	ogbl-citation2 |
> |:------:|:------:|:-------:|:-------:|:-------:|:-------:|:-------:|:-------:|
> |Node2Vec|	(32G, 1h)|	(32G, 1h)	|(32G, 1h)	|(32G, 4h)	|(32G, 4h)	|(32G, 4h)|	(32G, 6h)
> |MLP|	(32G, 1h)|	(32G, 1h)	|(32G, 1h)	|(32G, 4h)|	N/A	|(32G, 4h)|	(32G, 6h)
> |MF|	(32G, 1h)|	(32G, 1h)	|(32G, 1h)	|(32G, 4h)	|(32G, 4h)	|(32G, 4h)|	(32G, 6h)
> |GCN|	 (32G, 1h)|	(32G, 1h)|	(32G, 1h)	|(32G, 4h)|	(32G, 4h)|	**(32G, 24h)**|	(32G, 6h)
> |GAT|	 (32G, 1h)|	(32G, 1h)|	(32G, 1h)	|(32G, 4h)|	(32G, 4h)|OOM|	OOM
> |SAGE|	 (32G, 1h)|	(32G, 1h)|	(32G, 1h)	|(32G, 4h)|	(32G, 4h)|	**(32G, 24h)**|	(32G, 6h)
> |GAE|	 (32G, 1h)|	(32G, 1h)|	(32G, 1h)	|(32G, 4h)|	(32G, 4h)|	OOM|OOM
> |SEAL|	(32G, 1h)|	(32G, 1h)	| **(32G, 24h)**|	(32G, 4h)|	**(32G, 12h)**	|**(32G, 48h)**|	**(32G, 48h)**
> |BUDDY|	(32G, 1h)	|(32G, 1h)	|(32G, 1h)	|(32G, 1h)	|(32G, 1h)	|(32G, 4h)|	(32G, 10h)
> |NeoGNN	|(32G, 1h)|	(32G, 1h)|	(32G, 1h)|	(32G, 4h)|	(32G, 4h)|	(32G, 6h)|	**(48G, 10h)**
> |NCN|	(32G, 1h)|	(32G, 1h)|	(32G, 1h)|	(32G, 1h)|	(32G, 1h)|	(32G, 10h)	|**(48G, 4h)**
> |NCNC|	(32G, 1h)|	(32G, 1h)|	(32G, 1h)|	**(48G, 1h)**|	**(48G, 48h)**|	**(48G, 48h)**|	**(48G, 24h)**
> |NBFNet|	(32G, 1h)|	(32G, 1h) |	**(32G, 20h)**	|OOM|	>24h/epoch|	OOM|	OOM
> |PEG|	(32G, 1h)|	(32G, 1h)|	(32G, 1h)|	(32G, 1h)|	(32G, 1h)|	OOM	|OOM

---

> > ### Author Response · Authors · 2023-08-12
> > **Response of Raised Questions [Part 2/3]**
> >
> > > Q2: The set of heuristics chosen for the HeaRT selection is ad-hoc. Do the results depend on the choice of heuristics? If so, how is the current selection justified?
> >
> > The current set of heuristics is justified  by our interest in using multiple 'representative' link prediction heuristics that cover a wide variety of information. As a reminder, we restrict the negative samples to the corruptions of the corresponding positive sample. A simple strategy to obtain negative samples from the corruptions is through uniform sampling. However, this may yield 'easy' negative samples that are irrelevant to the specific nodes, thus making the task easily handled by many of the methods. In this case, the evaluation task may not be able to accurately reflect the performance of different methods. Hence, as inspired by the candidate generation process in real-world recommender systems [39, 40], **we aim to select a set of 'hard' negative samples that are more relevant to the target.** The candidate generation process is typically based on some primitive and simple link prediction heuristics. These  heuristics can be also treated as link prediction methods. Their performance is just passable and hence they are suitable to generate the set of 'hard' negative samples.
> >
> > In our case, the set of heuristics was chosen based on a careful review of literature. Most link prediction heuristics can be categorized into two main categories: **local heuristics** and **global heuristics** (``Link prediction in complex networks: A survey" [1]). Local heuristics attempt to capture the local neighborhood information that exists near the node pair while global heuristics attempt to use the whole graph structure. Note that these heuristics can be also treated as link prediction methods and often achieve reasonable performance for link prediction.
> >
> > Based on the existing heuristics, we aim to choose one representative from each category. For the  **local heuristic** we use Resource Allocation (RA), which considers the common neighbor information. Existing results show that RA can achieve strong performance on most datasets (see Tables 1 and 2). For the  **global heuristic** we use Personalized Pagerank (PPR), which considers the random walk information over the whole graph. Twitter and Pinterest (see [39, 40]) have used PPR (and other random walk based methods) for candidate generation in the past. Lastly, we further include the cosine feature similarity for Cora, Citeseer, and Pubmed. This is due to the relatively strong performance of an MLP on those datasets. By combining these heuristics, we are able to generate a diverse set of negatives for each positive sample.
> >
> >
> > We further analyze the sensitivity of the generated negative samples to the heuristics on the Pubmed dataset. **We first observe that changing only one type of heuristic has a small effect on the generated negatives.** We demonstrate this by first replacing the local heuristic with CN instead of RA. The negatives are then generated in conjunction with PPR and the feature similarity. We find that 80\% of the negative samples are the same with those that are use RA. This indicates a modest difference when changing the metric for any one type of information.
> >
> > Next, we analyze how the generated negative samples differ by each individual type of heuristics. We show that there is much variation between the generated negatives of different categories. This is demonstrated by first calculating the negatives produced by each of three heuristics, **individually** (e.g., only RA). We then calculate the overlap of those produced negatives. On Pubmed, we observe that only 3.5\% of the negatives are shared across each. This is as expected as the different types of heuristics capture much different types of information, enabling HeaRT to capture a wide range of different information.

---

> > > ### Author Response · Authors · 2023-08-12
> > > **Response of Raised Questions [Part 3/3]**
> > >
> > > > Q3: Beyond that, it is not clear to me that the authors publish their selected negatives. It's suggested to publish dataset to Zenodo.
> > >
> > >
> > > We have published the negative samples produced by HeaRT. They can be found [here](https://cse.msu.edu/~shomerha/HeaRT-Data/). We also detail how to download the samples in our Github Readme.
> > > Furthermore,  we appreciate the suggestion of using Zenodo and have published the dataset there as well. See [here](https://zenodo.org/record/8226022) for more details.
> > >
> > > > Q4: Citation [42] seems to have been included in error.
> > >
> > > Citation [42] (44 in the revised version due to the inclusion of other citations) is actually cited in Appendix C.2. It refers to the Adam optimizer, which is used throughout our experiments.
> > >
> > >
> > > > Q5: I'm not sure where the a in HeaRT comes from.
> > >
> > > We have named our model as 'HeaRT', an acronym derived from 'Heuristic Related Sampling Technique'. The inclusion of the letter 'a' in 'HeaRT' does not correspond to a specific word in the title; rather, it was deliberately added to enhance the memorability of the name and to provide a phonetic resonance with the word 'heart', This stylistic choice is aimed at fostering both recall and recognition within the context of our work.
> > >
> > > > Q6: Various grammar and writing suggestions.
> > >
> > > We appreciate all other comments made in regards to our writing and grammar. We greatly appreciate your help in improving the readability of our paper. All comments have been addressed in the revision.

---

> > ### Comment · Reviewer_vjFt · 2023-08-29
> >
> > > Assuming we consider sets of hyperparameters for each method, this would result in a total of different combinations. Even for a modest this can easily be prohibitively expensive.
> >
> > This seems to imply that you are using a full grid search on the parameters. I would say that this is not necessary. So long as each method is given comparable resources for hyperparameter tuning, a Bayesian/random search (as done in [8]) is sufficient. HOwever, both approach are fine, so this does not detract from the paper.
> >
> > > Lastly, we note that since [8] focuses on KGC and ours on link prediction we share no overlap in datasets nor methods.
> >
> > This is not quite true. KGC is nothing but link prediction on knowledge graphs, so the aims of the two papers are very comparable (although the datasets and exact methods are indeed different).
> >
> > The key difference, however, appears to be that in this paper you include message passing methods like GCNs, which are notoriously memory hungry, and potentially quite slow. As such, I'm satisfied that the requirements are different in this case.
> >
> > > Citation [42] (44 in the revised version due to the inclusion of other citations) is actually cited in Appendix C.2. It refers to the Adam optimizer, which is used throughout our experiments.
> >
> > Perhaps I made a typo in the number. I was referring to [45] in the current version, which reads "JC de Borda. M’emoire sur les’ elections au scrutin. Histoire de l’Acad’emie Royale des Sciences, 1781."
> >
> > Beyond that I am satisfied with the responses. I urge the authors to clarify these points in the paper as well as in their answers to me.

---

> > > ### Author Response · Authors · 2023-08-29
> > > **Response of new comments**
> > >
> > > > This seems to imply that you are using a full grid search on the parameters.
> > >
> > > Yes, we use a full grid search across all datasets to ensure the models have the same search space. This is a common approach among benchmark papers (see [6, 7, 11]).
> > > Thanks for pointing this out. We have updated our paper on line 137 to make this clearer.
> > >
> > > >  This is not quite true. KGC is nothing but link prediction on knowledge graphs, so the aims of the two papers are very comparable (although the datasets and exact methods are indeed different). The key difference, however, appears to be that in this paper you include message passing methods like GCNs, which are notoriously memory hungry, and potentially quite slow.
> > >
> > > We apologize for the confusion. We agree with the reviewer that KGC and link prediction are similar tasks. We further agree that the models used in our study require much more computational resources than those in [8].
> > >
> > > >  Perhaps I made a typo in the number. I was referring to [45] in the current version, which reads "JC de Borda. M’emoire sur les’ elections au scrutin. Histoire de l’Acad’emie Royale des Sciences, 1781."
> > >
> > > This citation is referenced on line 681 in the Appendix in the revision. It refers to Borda’s method, a method used for combining to ranked lists.
> > >
> > > > I urge the authors to clarify these points in the paper as well as in their answers to me
> > >
> > > Thank you very much for all your helpful feedback. We have revised our paper to make these points clearer. Specifically, we've include our clarification of the chosen heuristics in Appendix G.1 and the use of a grid search on line 137.

---

> ### Author Response · Authors · 2023-08-17
>
> Thank you again for taking the time to review our paper.
>
> We are writing to check if you have any further concerns or questions after reading our rebuttal. Since these may require extensive experiments or revisions, we want to ensure that we have sufficient time to address any concerns the reviewers may have.
>
> Thank you!

---

### Decision · Program_Chairs · 2023-09-22

**Decision:**

Accept (Poster)

**Comment:**

Paper addresses an important task (link prediction) with a variety of applications. Classifying a positive edge VS random is an easy task. However, in reality, e.g., in recommender systems, the model should be able to rank positives (few) over N^2 negatives (can be in the millions or billions). This paper bridges some of that gap, by creating a more-realistic link prediction evaluation benchmark with harder negatives.